# On Non-interactive Evaluation of Animal Communication Translators

## Abstract

If you had an AI Whale-to-English translator, how could you validate whether or not it is working? Does one need to interact with the animals or rely on grounded observations such as temperature? We provide theoretical and proof-of-concept experimental evidence suggesting that interaction and even observations may not be necessary for sufficiently complex languages. One may be able to evaluate translators solely by their English outputs, offering potential advantages in terms of safety, ethics, and cost. This is an instance of machine translation quality evaluation (MTQE) without any reference translations available. A key challenge is identifying "hallucinations," false translations which may appear fluent and plausible. We propose using segment-by-segment translation together with the classic NLP shuffle test to evaluate translators. The idea is to translate animal communication, turn by turn, and evaluate how often the resulting translations make more sense in order than permuted. Proof-of-concept experiments on data-scarce human languages and constructed languages demonstrate the potential utility of this evaluation methodology. These human-language experiments serve solely to validate our reference-free metric under data scarcity. It is found to correlate highly with a standard evaluation based on reference translations, which are available in our experiments. We also perform a theoretical analysis suggesting that interaction may not be necessary nor efficient in the early stages of learning to translate.

## 1 Introduction

Due to advances in language models (LMs), recent interest has surged in translating non-human animal communication, despite the complete lack of parallel training data (Goldwasser et al., 2023; Rodríguez-Garavito et al., 2025; Sharma et al., 2024b; Paradise et al., 2025). What type of data and interaction are required to validate translators? Evaluation is a crucial component of developing and validating such translators. An evaluation for reference-free translators may also prove useful in training translators, as we analyze.

The experiments and analysis in this paper suggest that for complex communication systems, we may be able to evaluate translators from their English outputs without any training data or grounded observations, such as temperature. At first, this seems impossible: a candidate English translation may be a total hallucination, appearing perfectly fluent. And indeed, LMs sometimes hallucinate entire translations, especially in low-resource languages where there is little training data. Figure 2 illustrates the difficulty of detecting a hallucinated translation without a reference translation.

A translator takes as input a communication in some format, e.g., raw audio or transcription (Sharma et al., 2024a; Hagiwara et al., 2024), and outputs text in a human language.[1] Our proposal is to:

- Segment each source communication by *turn*, i.e., which animal is vocalizing.
- Run the translator turn-by-turn.
- Judge how often the resulting translations make more sense in order than permuted.

We refer to this evaluation approach as *ShufflEval*. It can be viewed as an adaptation of of the classic shuffle test (Barzilay & Lapata, 2008; Laban et al., 2021; Iyyer et al., 2015; Taware et al., 2022), an

---

[1]Throughout this work outputs will be in English, though any human language could be used.

established approach in NLP, to the problem of Machine Translation Quality Evaluation (MTQE), specifically to Reference Free Quality Evaluation (RFQE) where there are no reference translations for comparison. We use reference-free to mean no reference translations; most MTQE uses source text, whereas our metric is target-only (it uses only translated segments plus their temporal order).

ShufflEval yields a score between 0 and 1, with a score above 0.5 (random guessing) being some degree of validation. ShufflEval is conservative in the sense that it may yield a score of 0.5 to a perfect translator of a simple source communication system where different segments have no semantic interdependencies. However, a ShufflEval score of greater than 0.5 means that shuffling translated segments decreases coherence, which would not be achieved by pure hallucinated translations. Figure 1 illustrates hypothetical translations of animal communication where a shuffle test may succeed or fail, even for perfect translation. Other possible failure modes are discussed in Section 4.

Of course, ShufflEval can be applied to any translator, including ones that translate human languages paragraph-by-paragraph. Until recently, it would have been difficult to run ShufflEval without human judges, because older LMs struggled to distinguish the original order of translated paragraphs from a permutation (Laban et al., 2021). This is no longer the case with modern reasoning models.

Since there is currently inadequate animal data to translate complex animal communication systems, we thus validate ShufflEval methodology on languages for which we do have reference translations, including low-resource languages and constructed languages (conlangs). We recognize that real human languages, regardless of data availability, are rich, complex, and valuable (Cooper et al., 2024). Our use here is strictly methodological to test ShufflEval in settings where references exist. We do not draw any conceptual similarity between these languages and animal communication. However, experiments on human language fail to capture the domain gap between animal communication and human communication. To mimic this enormous domain gap, we also perform experiments on ten artificial conlangs, generated using GPT-5 to be quite different from human languages.

ShufflEval addresses a weakness in prior RFQE techniques which is problematic especially when the source is poorly understood—namely that hallucinations may score as high as faithful translations (see Figure 2 for an example). Prior RFQE methods often evaluate the coherence and other qualities of the translation, e.g., by simply prompting an LM to score the quality of the English translation alone. Relatedly, there is growing work on hallucination detection for MT using LMs (Benkirane et al., 2024). These hallucinations can be coherent and score high on RFQE because they do not face the faithfulness/coherence tradeoff inherent in translation. ShufflEval complements these other RFQEs in that it is more resilient to hallucinations, but it is not a replacement. For example, it may be insensitive to AlTeRnAtInG cAsE, which impedes readability without significantly altering meaning. Thus, ShufflEval can be combined with other RFQE metrics.

When ShufflEval is inadequate (which may be especially likely for rudimentary animal signals) one may incorporate observational grounding, such as animal locations, temperature, feeding patterns, and even video. In Section 2.1, we model this with a loss $\ell(T, Z) \in [0, 1]$ that measures likelihood of translation $T$ with respect to grounding $Z$. Learning may be cast as minimizing the expected loss $\mathbb{E}[\ell(f(S), Z)]$ over translators $f \in \mathcal{F}$ in some family $\mathcal{F}$. ShufflEval is a special case of an observation-free loss (i.e., $Z = \emptyset$), and when $\mathcal{F}$ is restricted to segment-based translators.

**Ethical considerations.** Validation in animal studies often relies on interactive methods such as playback experiments (see e.g. Dabelsteen & McGregor 1996), in which sounds are played back to animals. These may cause welfare and ecological harms. This work suggests that playback may be avoided by enabling non-interactive evaluation (see Ethics Statement for discussion).

**Theoretical analysis.** We also present a theoretical analysis of the value of interaction in training a family of parametrized translators, leveraging the connection between evaluation and training. Corollary 2 argues that noninteractive evaluations use significantly fewer resources than interactive evaluations at the price of slightly worse translations. Quality is measured by a general bounded loss function for translations and compares its loss after training using interaction versus from observations alone (without interaction), based on certain quantities. The loss can depend on animal-specific observations, or in the case of the shuffle-eval it uses only temporal grounding and segment-level translators. This analysis goes beyond prior work in the literature (Goldwasser et al., 2023) which analyzes conditions under which unsupervised translation should be possible but does not provide any evaluation methodology. Limitations of this analysis are discussed in Section 4.

| | |
|---|---|
| **Whale A:** "Mother, let's go for a dive soon." | **Cow A:** "I'm here. Where are you?" |
| **Whale B:** "Little follower-of-my-echo, we just filled | **Cow B:** "I'm here. Where are you?" |
| our bellies. We only surfaced ten breaths ago." | **Cow C:** "I'm hungry." |
| **Whale A:** "It was fifty breaths." | **Cow A:** "I'm here. Where are you?" |
| **Whale B:** "Ten more breaths, and we will." | **Cow B:** "I'm here. Where are you?" |

Figure 1: Hypothetical translations which the shuffle test should succeed (left, message permutations are less plausible than the original order) and fail (right).

Source (in Santali):
ᱯᱟᱱᱮᱱ-ᱯᱟᱜᱟᱱ ᱜᱟᱴᱷ ᱯᱟᱜᱟᱴ ᱯᱚ ᱩᱟᱱᱮᱱ ᱩᱚ'ᱵᱟᱴ ᱟᱱ ᱯᱟᱜᱟᱱ ᱜᱟᱴᱷ ᱯᱮᱯᱩᱫ ᱠᱩᱱᱠᱟᱴ ᱯᱚ ᱠᱟᱭᱫᱚᱟᱜᱟᱱ, ᱠᱚᱣᱚᱱ ᱩᱟᱱᱮᱱ ᱟᱫ ᱩᱟᱯ ᱯᱮᱯᱩᱫ ᱯᱟᱜᱟᱴ ᱚᱟᱣᱫ' ᱵᱟᱱᱮᱱ ᱤ ᱮᱚᱩᱟ ᱯᱚ ᱯᱮᱮᱮ ᱠᱟᱯ ᱟᱫ ᱚᱟᱱᱷᱟᱠ ᱵᱚ ᱩᱮᱮᱟᱱ ᱯᱮᱯ ᱚᱟᱣᱮ'ᱯ. ᱟᱱ ᱮᱚᱩᱟ ᱯᱚ ᱚᱟᱯᱟ ᱚᱟᱯᱟ ᱚᱮ ᱯᱟᱯᱟᱱ ᱯᱮᱮᱱ ᱤ ᱮᱚᱩᱟ ᱯᱮᱯᱩᱫ...

**Translation A**: *The Sarna-Liturgical Kanthi (rosary) of the Sarna faith, the independent religious system of the indigenous people, and the sacred text of Sarna-Liturgical came to light in the light of detailed research, conversation, and meticulous efforts. According to the 2011 census, Sarna is a believer of 46,73,848 people in the country, and he is recognized as a...*

**Translation B**: *Patna–Digha Ghat line is a railway line between Patna Junction and Digha Ghat railway stations in the city of Patna, Bihar. It was built by the British in 1862, and later trains were operated on it. In 1962–63 and again in 2004 the Railway Ministry, and Lalu Prasad Yadav, took steps regarding this line. However, regular passenger train services...*

**Translation C**: *The Wild Animals and Their Association*
*By the tender age of four or five, when our bodies are neither weak nor strong, the adivasi children probably learned to live with bravery and dignity without even feeling it. Indeed, we can suspect they gain strength from the innocent aspects of wild animals and perhaps have...*

Figure 2: Which translation is *not* a hallucination? Without a reference translation, one cannot distinguish faithful translations from hallucinations. See Appendix E.1 for the answer.

## 2 THEORETICAL ANALYSIS

We now present an analysis suggesting that in the high-error regime (i.e., for translators with relatively low accuracy), observations may be more cost effective than interactive experiments. It is instructive to consider interaction in supervised learning and how it relates to unsupervised machine translation. We defer this comparison the the appendix, Appendix C.

### 2.1 TRANSLATOR AND LOSS DEFINITIONS

We use $\log_2$ (resp. $\ln$) to denote logarithms in base 2 (resp. $e$). Let $\mathcal{F}$ be a family of **translators** $f : \mathcal{S} \to \mathcal{T}$ from **sources** $S \in \mathcal{S}$ to **targets** $T \in \mathcal{T}$. This paper is motivated by the setting that $\mathcal{S}$ consists of animal communication and $\mathcal{T}$ is English translations, but our theory holds generally.

For simplicity, we assume translators are deterministic (though the analysis remains essentially unchanged for randomized translators) and $\mathcal{F}$ is finite; then, informally, the number of "essential parameters" is proportional to $\log_2 |\mathcal{F}|$, i.e., the number of bits required to represent a translator.[2] There is an arbitrary joint distribution $(S, Z) \sim \mathcal{D}$ over sources and **observations** $Z \in \mathcal{Z}$, e.g., geoposition or water temperature measurements.

Performance is evaluated by a bounded *loss* $\ell(T, Z) \in [0, 1]$ measuring how unlikely a translation $T$ is with respect to observation $Z \in \mathcal{Z}$. The optimal loss for translator family $\mathcal{F}$ is

$$\mathrm{opt} := \min_{f \in \mathcal{F}} \ell_{\mathcal{D}}(f) \text{ where } \ell_{\mathcal{D}}(f) := \mathop{\mathbb{E}}_{(S,Z) \sim \mathcal{D}} \big[ \ell(f(S), Z) \big]. \tag{1}$$

We assume that the per-sample loss $\ell(\cdot, \cdot)$ can be efficiently evaluated, and therefore gives an actionable metric for measuring translation quality.

---

[2]In practice, many classes are "over-parametrized" meaning that smaller models (e.g., sparsified models or families of smaller size) can achieve similar performance. For simplicity of analysis, we assume that model family is already compressed.

Our analysis proceeds in two parts. In Section 2.2 we show how a standard learning-theoretical scaling law can be applied to translation from observations. Section 2.3 compares this analysis to a *whalebreak* scaling law for learning with interaction, revealing that in the low-accuracy regime, observations may suffice for learning—and at cheaper monetary and ethical cost.

## 2.2 LEARNING TO TRANSLATE FROM OBSERVATIONS

Our notion of an accurate translator is based on low average loss $\ell(T, Z) \in [0, 1]$ which can be evaluated efficiently on any translation $T \in \mathcal{T}$ and whatever observations $Z \in \mathcal{Z}$ are available, if any. We consider the translator learning algorithm which selects the translator $\hat{f} \in \mathcal{F}$ (i.e., chooses its parameters) so as to minimize the average empirical loss on $m$ training data $\langle S_i, Z_i \rangle_{i=1}^m$, i.e.,

$$\hat{f} := \operatorname*{argmin}_{f \in \mathcal{F}} \hat{\ell}(f), \quad \text{where } \hat{\ell}(f) := \frac{1}{m} \sum_{i=1}^m \ell(f(S_i), Z_i). \tag{2}$$

In the case of multiple translators with equal average empirical loss, we assume some arbitrary tie-breaking procedure. Of course this algorithm is computationally infeasible, but this approach (often called Empirical Risk Minimization (Kearns & Vazirani, 1994)) is common in statistical learning theory.

We now observe that $\ell_{\mathcal{D}}(\hat{f})$ converges at a rate of $O\left(\sqrt{\frac{1}{m} \log |\mathcal{F}|}\right)$ where again $\log_2 |\mathcal{F}|$ is the compressed model bit-size, proportional to the number of essential parameters.

**Theorem 1** (Observational scaling law). *Let $\mathcal{F}$ be a finite set of translators $f : \mathcal{S} \to \mathcal{T}$, $\mathcal{D}$ be an arbitrary distribution over $\mathcal{S} \times \mathcal{Z}$, $\delta \in (0, 1)$, and $\ell : \mathcal{T} \times \mathcal{Z} \to [0, 1]$ be a bounded loss. With probability $\geq 1 - \delta$ over $m$ i.i.d. samples $(S_i, Z_i) \sim_{i.i.d.} \mathcal{D}$, the translator $\hat{f}$ which minimizes training loss satisfies,*

$$\ell_{\mathcal{D}}(\hat{f}) \leq \min_{f \in \mathcal{F}} \ell_{\mathcal{D}}(f) + \sqrt{\frac{2 \ln(|\mathcal{F}|/\delta)}{m}}, \tag{3}$$

*where $\hat{f}$ is defined in Eq. (2) and $\ell_{\mathcal{D}}$ is defined in Eq. (1).*

Setting $\delta = 0.01$, this means that with probability $\geq 99\%$ minimizing training loss yields $\hat{f}$ with true (population) risk $\ell_{\mathcal{D}}(\hat{f}) \leq \min_{f \in \mathcal{F}} \ell_{\mathcal{D}}(f) + \sqrt{\frac{10 + 2 \ln |\mathcal{F}|}{m}}$. All proofs are deferred to Appendix F.

## 2.3 COMPARING OBSERVATIONAL AND INTERACTIVE LEARNING

To compare observational learning to interactive learning, we consider a simple optimistic model of interactive experiments. Suppose that $n$ interactive experiments are used to find the best classifier $f_n^\star \in \mathcal{F}_n$ in terms of minimal expected loss $\operatorname{opt}_n := \min_{f \in \mathcal{F}_n} \ell_{\mathcal{D}}(f)$. With more experiments $n$, one may be able to learn larger families $\mathcal{F}_n$, hence the translator family grows with $n$. We call this the "whalebreak" model, alluding to "jailbreaks" which remove restrictions on language models, sometimes revealing internal details word-for-word such as system messages (Zhang et al., 2023).

Why is this an *optimistic* model? Let $b = \frac{1}{n} \log_2 |\mathcal{F}_n|$. Since $\log_2 |\mathcal{F}_n|$ is the number of bits required to encode a translator, $b$ can be interpreted as the average number of parameter bits learned per experiment. In the extreme case of binary search, where each experiment rules out half of the remaining hypotheses, $b = 1$. In noisy or less structured interactive settings, one expects $b \ll 1$. Thus the whalebreak model effectively grants interactive experiments maximal information efficiency, as though each query reveals the next "bit" of the optimal translator. This framing is deliberate: as we show next, observational data can in fact be competitive with interaction even under this generous assumption—and at a fraction of the cost. Hence, the case for observational data is only stronger in more realistic scenarios.

**Setup: Experiment costs and budget reduction.** The above stylized model enables a simple comparison to observational learning. The key assumption we need is that observations are less expensive in terms of the relevant *cost*, which we treat as an abstract quantity. Cost can capture monetary expense, but also the time and effort of human participants, or even the ethical burden of interfering with wildlife (as is sometimes the case in interactive playback experiments). Formally,

we assume that observing a fresh iid sample $(S, Z) \sim \mathcal{D}$ costs at most an $\varepsilon$-fraction of the price of conducting a single interactive experiment, for some *cost ratio* $\varepsilon \in (0, 1)$. The goal is a reduced total budget: we allow the observational learner to spend only a $1/c$-fraction of the interactive cost, for some *budget reduction factor* $c > 1$.

**Corollary 2.** For any cost ratio $\varepsilon \in (0, 1)$ and budget reduction factor $c > 1$, suppose that $n$ interactive experiments identify a translator of minimal loss $\mathrm{opt}_n \coloneqq \min_{f \in \mathcal{F}_n} \ell_{\mathcal{D}}(f)$. Then, at only a $1/c$-fraction of the interactive cost, the translator learned from observations $\hat{f}_n \coloneqq \mathrm{argmin}_{f \in \mathcal{F}_n} \hat{\ell}(f)$ will, with probability at least $0.99$, satisfy

$$\ell_{\mathcal{D}}(\hat{f}_n) \leq \ell_{\mathcal{D}}(f_n^\star) + \sqrt{\varepsilon c \left( \frac{3b}{2} + \frac{10}{n} \right)},$$

where $b \coloneqq \frac{1}{n} \log |\mathcal{F}_n|$.

See Appendix F for the proof. If, as in the previous section, we assume $\log_2 |\mathcal{F}_n| \leq n$, then setting $c = 4$ shows that at just one quarter of the interactive budget, the observational model achieves loss within $\sqrt{\varepsilon (6 + 40/n)}$ of the interactive model (with 99% likelihood). For example, when $\varepsilon = 1/2400$, this gap is at most $\sqrt{6\varepsilon} \approx 0.05$. In the current state of animal communication research, even coarse-grained understanding remains elusive, so the relevant regime is one of relatively large losses. In such a regime, the bound indicates that observational data can be nearly as effective as interactive experiments, while incurring only a fraction of the cost. If, at some future stage, accuracies above (say) 0.9 become achievable, then the tradeoff may shift and interactive experiments could become valuable. But at present, when even rough translation is beyond our reach, the analysis supports the view that observational approaches are both cost-effective and scientifically sufficient.

### 2.4 IMPLICATIONS FOR SHUFFLEVAL

Corollary 2 implies that, in theory, ShufflEval can be used to non-trivially evaluate translator quality (e.g. as compared to the supervised or even interactive settings). Before we validate this theory in Section 3, it may be enlightening to elaborate this implication in more formal detail.

The starting point of ShufflEval is a segment-by-segment translator. To capture this formally, we assume that each source communication is partitioned into $k \geq 2$ segments $S = s_1 s_2 \cdots s_k$. As explained in the introduction, these segments may correspond to different "turns" in a dialogue, or even simply to a temporal partition of an audio recording. ShufflEval evaluates segment-by-segment translators. Therefore, we assume that all translators $f \in \mathcal{F}$ satisfy $f(s_1 \cdots s_k) = \varphi(s_1) \cdots \varphi(s_k)$ for some segment-translator $\varphi \colon s \mapsto t$. To avoid cumbersome notation, we will refer only to the translator $f$ from here onwards. Furthermore, we assume without loss of generality that any target communication $T$ is segmented into $T = t_1 \cdots t_k$ (e.g. by inserting a special symbol between each segment in the translation $f(S)$).

The final component of ShufflEval is a *plausibility (pairwise-preference) model* $\rho \colon \mathcal{T} \times \mathcal{T} \to \{0, 1\}$; here $\rho(T, T') = 1$ indicates that $T'$ is more plausible than $T$, and $\rho(T, T') = 0$ indicates $T$ is more plausible than $T'$. (In Section 3 we show how such $\rho$ can be obtained from an LLM.)

In experiments, we consider the ShufflEval score $1 - \ell_{\mathrm{ShufflEval}}(T)$ which is better when larger. The ShufflEval loss of a translation $T = t_1 \cdots t_k$ is defined to be

$$\ell_{\mathrm{ShufflEval}}(T) \coloneqq \frac{1}{k! - 1} \sum_{\pi \in \Pi_k \setminus \{\mathrm{Id}\}} \rho \left( t_1 \cdots t_k, \, t_{\pi(1)} \cdots t_{\pi(k)} \right),$$

where $\Pi_k$ denotes the set of all permutations $\pi \colon [k] \to [k]$, and $\mathrm{Id}$ is the identity function. Put simply, ShufflEval measures whether permuting the outputs of the translator affect plausibility; intuitively, assuming that order matters in the source, then an accurate translator should output translations whose internal order matters as well, i.e., $\rho(t_1 \cdots t_k, t_{\pi(1)} \cdots t_{\pi(k)}) = 0$ for most $\pi$.

We note that computing ShufflEval naively requires exponentially (in $k$) many invocations of the plausibility model $\rho(\cdot, \cdot)$. In practice, for large $k$ the loss can be approximated by sampling random permutations $\pi \in \Pi_k$ and reporting the empirical mean (or using importance sampling).

ShufflEval is a special case of Corollary 2, by taking $\ell(T, Z) = \ell_{\mathrm{ShufflEval}}(T)$. At first glance, it may seem that ShufflEval does not depend on any grounding (observations $Z \in \mathcal{Z}$). However, there is an

```
ShufflEval prompt

We have a (possibly poor) English translation of source_description,
broken into segments. To make matters worse, we are not certain what
order the segments should be in.

Below are two orderings of the segments.
Decide which ordering reads more natural and coherent.
Reply with '1' or '2' only.

<ORDERING1>
text1
</ORDERING1>

<ORDERING2>
text2
</ORDERING2>
```

Figure 3: Our template for ShufflEval. Either `text1` or `text2` is permuted and the other is in the original order. Each of ten permutations is run twice, swapping order to account for order bias.

implicit common ground: *time*. The sequential representation which imposes an order on segments implicitly assumes a temporal order to the communication, so that it can be partitioned in a way that can be translated segment-by-segment. This would not hold if there was no common linear order between the source and the translation (e.g. images). Put another way, it is not clear how to apply ShufflEval to image captioning, but it could be applied to describing videos. Augmenting ShufflEval to combine both temporal consistency and observations $Z \in \mathcal{Z}$ is deferred to future work.

## 3 EXPERIMENTS

We validated the shuffle test as a viable RFQE signal, evaluating fifteen LMs from the OpenAI API as translators of conlangs and low-resource Wikipedias. The target language for all these experiments is English. We use LMs for several purposes, many of which follow increasingly common practices in MT (Bavaresco et al., 2025). These include:

- **Translators.** We used fifteen LMs accessed via the OpenAI API as translators. The prompt for translating a source is shown in Figure 8. The LMs are numbered and listed in Figure 5.
- **ShufflEval judge.** GPT-5 (LM 15) was used to judge which of two orderings is more plausible, with the prompt of Figure 3, for reasons discussed below.
- **Baseline: reference-based MTQE judge.** GPT-4 (LM 4) has been repeatedly found to be consistent with human ratings of translation quality, with and without reference translations (e.g., Kocmi & Federmann, 2023; Liu et al., 2023; Jiang et al., 2024). We use GPT-4 with the prompt of Figure 4 to align with these prior studies.
- **Conlang creation.** GPT-5 is used to generate conlangs, as described in Section 3.3.

The LMs used are associated with numbers 1-15 based on their performance on the ground-truth shuffle test of Figure 5, which is not based on translation. All LMs were used with their default OpenAI API parameters. Code and data will be made available upon publication.

### 3.1 SHUFFLEVAL

For translation, we first independently translate the source segment-by-segment (paragraphs for Wikipedia articles, sentences for conlangs, and animal "speaker" turns might be used). We then perform pairwise comparisons using the prompt of Figure 3. To account for order bias, each comparison is performed twice with the two options swapped, and the results averaged. We average over ten random permutations, thus using twenty evaluations per (segment-level) translation.

```
Reference-based baseline prompt

Score the following translation on a continuous scale 0 to 100 where
score of zero means "no meaning preserved" and score of one hundred
means "perfect meaning and grammar".

<HUMAN_REFERENCE>
target
</HUMAN_REFERENCE>

<MACHINE_TRANSLATION>
translation
</MACHINE_TRANSLATION>

Just output an integer score between 0 and 100, inclusive, and
nothing else.
```

Figure 4: Prompt for judging translation quality with respect to a reference, closely following GEMBA-DA_ref from Kocmi & Federmann (2023) except without the source text.

As discussed, prior work (Barzilay & Lapata, 2008; Laban et al., 2021; Iyyer et al., 2015; Taware et al., 2022) has validated the shuffle test over sentences as a measure of coherence, finding that models specifically trained for the shuffle test were highly accurate at the sentence level, but struggled at longer segments such as paragraphs (Laban et al., 2021).

**Choosing a judge LM.** To compare the model performance on ShufflEval, we performed an experiment involving no translation. We took the paragraphs of English text and tested the models ability to distinguish the original order from permutations. The text comprised the first six paragraphs of 100 English Wikipedia articles (the same 100 English Wikipedia reference articles used in our low-resource experiments, described below). LMs were evaluated on accuracy at identifying the correct order the paragraph versus ten random alternatives.

As can be seen in Figure 5, newer models have better accuracy, with the earliest model, GPT-3.5 having only 61% accuracy, while model the latest model, GPT-5, achieved 96% accuracy. Random guessing would yield 50% accuracy. Order bias was present in several models, with models 2, 3, and 4 exhibiting preference for the first answer 60-70% of the time. The `gpt-4.1-nano -2025-04-14` model exhibited 99.9% order bias and was thus excluded from experiments.

### 3.2 LOW-RESOURCE HUMAN LANGUAGES (PROXY EXPERIMENTS)

We create a dataset from Wikipedia of 200 articles in data-scarce languages and their English versions, comprising 10 articles from each of 10 source languages and their English Wikipedia versions. High-quality parallel datasets only exist for some of these languages, e.g., Tamazight (Oktem et al., 2025). Therefore we use Wikipedia for a uniform approach.

**Sources.** Ten low-resource languages were selected by sorting the Wikipedias by how many entries they have, and from each taking ten articles that met the following criteria: (a) the article had at least 3,000 characters, (b) it was created after a cutoff date of June 1, 2024, and (c) it had an English version in Wikipedia. Languages which did not have 10 articles of this form were excluded. The English version was considered the reference-translation, which is a limitation discussed in Section 4. The ten languages can be seen in Figure 6 (right).

Articles were split into segments that were single paragraphs using GPT-4o. We took the first six paragraphs from each article. Limiting the number of paragraphs increased the alignment of the source and English versions of the articles, since the articles are not direct translations of each other.

Translations were performed one segment at a time, and the ShufflEval and baseline reference-based MQTE were both run as described above. We computed the correlation (Pearson correlation coefficient, in $[-1, 1]$) between the ShufflEval and the reference based score. Across the full

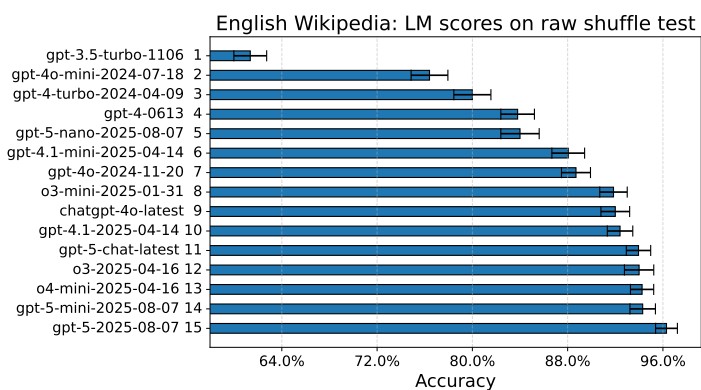

Figure 5: Without translation, accuracy at distinguishing the original paragraph order from a permutation. The fifteen LM's evaluated ranged from GPT-3.5-turbo (LM #1) to GPT-5 (LM #15).

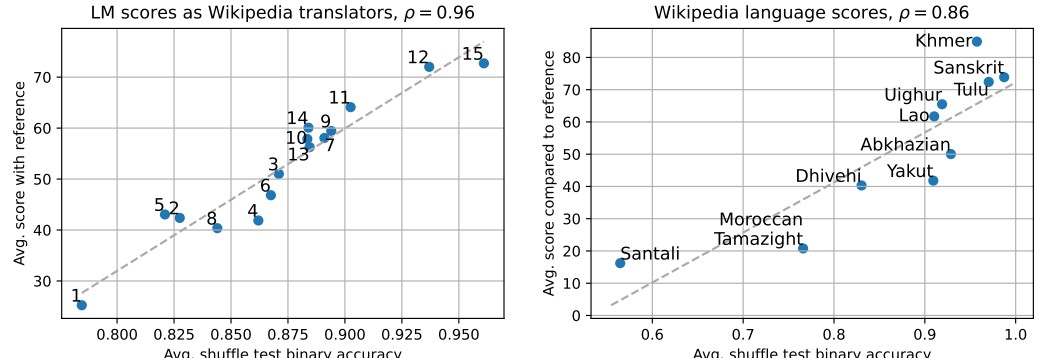

Figure 6: Comparing the reference-free ShufflEval (x-axis) to the reference-based baseline translation scores, averaged across LM (left) and language (right). On the left, LM numbers are taken from Figure 5 with 1 being GPT-3.5-turbo and 15 being GPT-5.

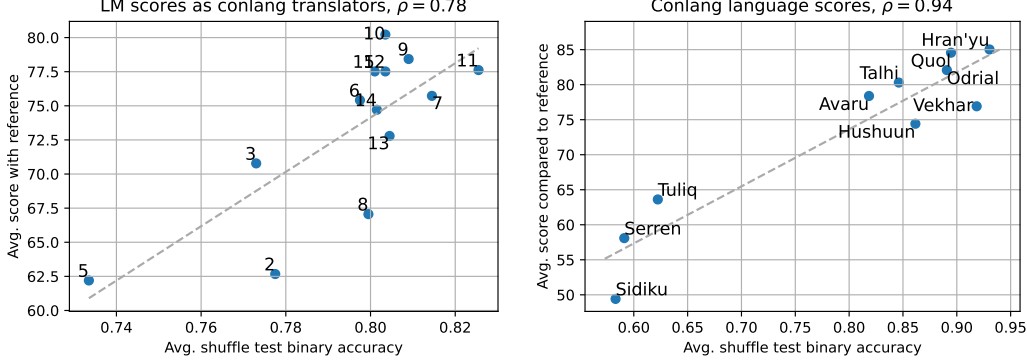

Figure 7: For conlangs, the reference-based baseline translation scores versus the reference-free ShufflEval, aggregated by LM (left) and language (right). LMs 1 and 4 were excluded due to inadequate context length.

$15 \times 100 = 1500$ pairs of articles and translators, the correlation was 0.54 ($\pm 0.04$ for 95% bootstrapped confidence interval) which is significantly positive. However, Figure 6 shows that when aggregated across models (left) and languages (right), ShufflEval is much more highly correlated. That is, we compute per-language and per-LM scores by averaging the two scores over those languages and LMs. One possible explanation for why the aggregate correlations are so much higher than per-article correlations is that the aggregate measure washes out multiple types of "noise" in the experiment. First, English versions of articles are sometimes not actual translations of the source document. Second, the baseline measure may be noisy and ShufflEval is run only with 10 permutations which introduces a certain amount of noise.

**Hallucination.** Translating an entire article, not in segments, runs the risk of hallucination. The common approach of direct RFQE without the shuffle test (e.g., Kocmi & Federmann, 2023) may evaluate a model that hallucinates fluently higher than a model that is better a translating (Zhang et al., 2024). Appendix E demonstrates this phenomenon in Wikipedia data.

### 3.3 CONSTRUCTED LANGUAGES

To stress-test ShufflEval beyond human languages, we ventured into constructed languages (conlangs) which permit the design of languages quite different from human languages. For example, the conlang Kēlen, spoken by the fictitious Kēleñi on the planet Tērjemar, has no verbs (Sotomayor, 1998). We used GPT-5 to construct ten diverse conlangs. These *artificial conlangs* were specifically synthesized to be quite different from human text, leveraging GPT-5's extensive knowledge of language and conlangs. Interestingly, in recent independent work, Alper et al. (2025) also constructed conlangs using LMs.

Each conlang includes a conculture, a detailed conlang definition, and ten sources alongside reference English translations. The creation pipeline is given in Appendix D.1, and a summary of the ten conlangs is in Table 1. While admittedly unrealistic, these conlangs do serve as tests where the domain gap between these conlangs and English is large, like animal communication.

The same experimental design was applied to both the Wikipedia and conlang settings, with two differences. First, the conlang segments were sentences rather than paragraphs, which offers a test of the ShufflEval at a different segmentation granularity. Second, in order to translate the conlang text, the conlang and conculture definitions were provided in the LM prompt, as seen in Figure 8 (bottom). Figure 7 shows the results comparing the baseline and ShufflEval on artificial conlangs.

**Discussion.** Across both Wikipedia articles in low-resource languages and artificial conlangs, there was a significant correlation between reference-based evaluations and ShufflEval's ranking of translators and languages. Of these comparisons, the translator scores (Wikipedia correlation 0.96, conlang correlation 0.78) reflects ShufflEval's potential to rank translators, which is relevant if one were to use such an metric during training. The language scores (Wikipedia correlation 0.86, conlang correlation 0.94) are more relevant to validation, where one may wish to compare the translations of an unknown animal communication system to other (possibly known) languages.

## 4 LIMITATIONS AND CONCLUSIONS

ShufflEval is far from perfect. First, it requires segmentation and translation of individual segments to be possible. Second, one can conceive of imperfect translators to which ShufflEval gives a perfect score, e.g., if segments started with increasing numbers, then a translator which only translates these numbers may be scored perfectly. Fortunately, such a flaw may readily be discovered by examining the target translations.

Multi-lingual Wikipedia articles are not direct translations of one another. However, non-translation references may be expected to decrease the correlation between ShufflEval scores and those determined using references. Thus, the strong correlations still serve as validation despite this issue.

The theoretical analysis also has several caveats. Exceptions to Corollary 2, where interaction may be particularly beneficial, include computational complexity, fine-grained delineations, out-of-distribution content, and counterfactuals.

Despite these limitations, this work presents a new approach to validate animal translation that benefits from less interaction, and opens the door to future improved tests.

**Ethics statement.** Better methods for understanding animal communication may have significant impact for science and conservation efforts. ShufflEval offers safety, ethics, and efficiency advantages over interactive evaluation; in the context of animal communication, interaction may come in the form of a *playback experiment* (see, e.g. Dabelsteen & McGregor 1996), in which sounds are played back to animals. Playback experiments raise ethical concerns because artificial signals can aggravate animals, trigger defensive or aggressive responses, and distort natural behavior. Across taxa, predator or conspecific-sound playbacks have disrupted vital activities—e.g., adult male sperm whales aborted foraging/resting dives and clustered socially in response to killer-whale calls (Curé et al., 2013), toadfish suppressed calling and showed elevated cortisol after dolphin-sound exposure (Remage-Healey et al., 2006), and chronic predator-noise exposure reduced song sparrow reproductive output by 40% via fear-mediated effects (Zanette et al., 2011). Playbacks can also create behaviors rather than merely reveal them, as daily affiliative call playbacks in marmosets transiently imposed a high-affiliation "cultural style" on groups (Watson et al., 2014), and the impacts of playback can persist for years or even a lifetime (Oñate-Casado et al., 2021). Given these welfare risks and ecological costs, we view the main ethical contribution of our work as demonstrating a path toward evaluating animal communication translators without playback.

We also recognize that many low-resource human languages are associated with marginalized communities. Our use of these languages is strictly methodological: they provide a controlled setting where ground-truth translations exist, allowing us to compare our reference-free metric against established evaluation methods. Low-resource languages were used in this paper because they are (by definition) data-scarce—and often underrepresented in the large language models used in our experiments—yet still come with parallel corpora that enables comparison to the gold-standard evaluation method (which uses references). Importantly, our use does not imply any similarity between real human languages and animal communication. We make this clear in the paper both explicitly (by stating their proxy role) and implicitly (by leading with constructed-language evaluations).

**Reproducibility statement.** All code and prompts needed to reproduce our results will be made available upon publication. The reproducibility relies on the availability of the API, however similar experiments may be run using our code on other reasoning LMs. In addition to the code, the experimental setup is described in Section 3 and in the Appendix. Finally, the formal specifications and necessary assumptions for the theoretical analysis are detailed in Section 2.

**Acknowledgements.** This study was funded by Project CETI via grants from Dalio Philanthropies and Ocean X; Sea Grape Foundation; Virgin Unite and Rosamund Zander/Hansjorg Wyss through The Audacious Project: a collaborative funding initiative housed at TED.

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

## A    Use of LMs in research and writing

Large language models were used extensively but carefully to aid in coding and writing the paper. LM suggestions were carefully reviewed, and detected errors were corrected. The LM outputs in the experiments, however, were analyzed numerically and reviewed mostly by spot checking. The authors examined conlang sources and translations and found that the *Talhi* conlang had source texts which included the translations within them, which were removed programmatically.

## B    Related work

The evaluation of translation quality without reference translations touches on several distinct research areas. While the literature is vast, we focus here on the most directly relevant connections to ShufflEval and its analysis.

**Local coherence in NLP.**    Works as early as that of Barzilay & Lee (2004) used random permutations of text as a method for empirically evaluating content (rather than translation) models. ShufflEval can be viewed as an adaptation of local coherence approaches in NLP, specifically the Shuffle Test introduced by Barzilay & Lapata (2008). Laban et al. (2021) demonstrated that supervised finetuning of a transformer can lead to near-perfect accuracy on the task, but performance drops significantly as the size of shuffled blocks increases.

**Reference-Free Machine Translation Quality Evaluation**    Reference-free evaluation has become increasingly critical for scenarios where obtaining references is impossible or expensive. Particularly notable is the reference-variant of COMET (Rei et al., 2022; 2020).

**Active Learning Theory**    Active learning theory reveals when interaction may not be necessary for learning. As surveyed by Hanneke (2014), active learning can achieve exponential improvements over passive learning under favorable conditions—specifically when complexity measures like the "star number" are finite (Hanneke & Yang, 2015). However, in high-noise regimes or for classes with infinite complexity, active learning may provide no advantage over passive learning. This theoretical insight suggests that for animal communication translation problems, passive evaluation through coherence assessment might suffice without requiring interactive feedback, particularly in early learning stages where noise levels are high. Our theory in Section 2 corroborates this intuition.

**Unsupervised Translation and Reference-Free Evaluation**    Unsupervised machine translation (Ravi & Knight, 2011) learns translators using only monolingual corpora from each language. While neural approaches have shown promise (Lample et al., 2018), empirical evaluations reveal that UMT is outperformed by supervised methods even with orders of magnitude more data (Kim et al., 2020). The key barriers identified include domain and data gaps. Recent theoretical work (Goldwasser et al., 2023) analyzes conditions under which UMT is possible without parallel data or shared domains, showing that translation feasibility depends on language complexity and common ground— suggesting animal communication translation may be possible if the system is sufficiently complex. Crucially, such a setting inherently requires reference-free evaluation, as parallel data is unavailable by definition. Our work complements this theoretical foundation by addressing the evaluation challenge: while Goldwasser et al. (2023) establishes when translation is possible, we provide a method to assess translation quality without references—a necessity for both unsupervised MT development and scenarios like hypothetical whale-to-English translation where references cannot exist.

**Constructed language for controlled experiments.**    As mentioned, in independent recent work, Alper et al. (2025) constructed conlangs using large language models. They also use a pipeline to construct languages, though their pipeline is designed to create more linguistically plausible languages involving consonants and vowels, phonology, morphology, and grammar rules. As a result, one might expect our conlangs to be less human-like which serves the purpose of stress-testing ShuffleEval beyond human languages. The widely popular BIG-bench (Srivastava et al., 2023) uses constructed language translation as one of its benchmarks, with similar "puzzles" appearing beforehand in NLP literature (Şahin et al., 2020).

## C INTERACTION IN SUPERVISED LEARNING

Interaction in machine learning has been extensively studied in supervised learning, particularly binary classification. It provides a helpful prelude to our analysis of interaction in translation. In standard binary classification, the goal is to learn a binary classifier $c : \mathcal{X} \to \{-, +\}$ using $m$ labeled samples $(x_i, y_i = c(x_i))$ for iid $x_i \sim \mathcal{D}$. *Active learning* adds interaction by allowing the learner to arbitrarily choose specific examples $\{x_j\}_j$ and see their labels $c(x_j)$—importantly, here the $x_j$ need not be distributed iid from $\mathcal{D}$.[3] In theory, active learning can provide exponential advantages for learning restricted families $\mathcal{C}$ of binary classifiers $c$ in noise-free settings, though these benefits typically do not extend to higher-dimensional or noisy problems (Hanneke, 2009).

**Binary search example.** The classic example is learning a binary threshold in one dimension, e.g., what counts as "warm" versus "cold" from examples of temperatures labeled $\pm$ based on whether the temperature $x > v$ is greater than some threshold $v$. Active learning enables one to select examples $x$ using binary search. For example, if 30 is labeled as $+$ and 10 is labeled as $-$, one queries 20, and if it is, say $+$, one then queries 25, and so forth. Binary search rapidly achieves *exponential* precision in the threshold $v$. On the other hand, given $m$ random labeled temperature samples $(x_i, y_i)$ where $y_i = \text{sgn}(x_i - v)$, it is not difficult to see that one can predict on future temperatures with error rate $\approx 1/m$ (the probability that a new example is closer to the threshold than any of the $m$ training examples).

Excitement from this exponential statistical savings was tempered by the realization that the savings does not generalize even to two-dimensional linear thresholds (Hanneke, 2009). Moreover, active learning has not proven popular in practice, e.g., common machine learning and statistical frameworks such as scikit-learn (Pedregosa et al., 2011), R (R Core Team, 2024), and TensorFlow (Abadi et al., 2016) do not have active learning modules.

One might hope to apply these active learning insights to translation by reducing it to binary classification: classify examples $x = (S, z)$ as positive when grounding data $z$ is consistent with source $S$ and negative otherwise. However, while source-grounding pairs from translation training data provide natural positive examples, it's unclear how to construct meaningful negative examples—what would constitute grounding data that's "inconsistent" with a source? This difficulty suggests that translation requires its own theoretical framework, which we develop in Section 2.

## D FURTHER EXPERIMENTAL DETAILS

### D.1 CREATING ARTIFICIAL CONLANGS

Ten artificial conlangs were generated through a series of prompts to GPT-5 (LM 15). Our conlangs are summarized in Table 1. We now describe the generation process. An initial attempt to generate a conlang in a single LM call failed in two ways: (1) the languages were too similar and lacked diversity, and (2) the descriptions were very short and incomplete. Therefore, we adopted a multistage generation pipeline.

In the first step, we elicit names for the conlangs, the species that speak them, the planets on which they live, as well as an unexpected property of the species and their communication. See Figure 9 (top). This encourages a diverse set of languages that differ significantly from human language. Ideating ten ideas in a single prompt led to more diversity than making ten different calls to the GPT-5. For example, when prompted separately, most planet names began with the letter X.

Second, like many human-authored conlangs, the artificial conlangs have an associated "conculture" that describes the fictitious speakers. We generate these vivid detailed descriptions, using the template of Figure 9 (bottom), based on the properties defined in the first step.

Third, we generate the conlang description itself, based on the conlang prompt of Figure 10.

---

[3]An alternative formulation is to give the learner access to an arbitrarily long sequence of samples $(x_j)_j$ from which they can choose which $x_j$'s to label. This equivalent formulation is less helpful as a warm-up.

```
Translate low-resource language prompt

Translate the following text from source_language to English:

<SOURCE_LANGUAGE>
text
</SOURCE_LANGUAGE>

Your output should be in the following format:

<ENGLISH_TRANSLATION>
...
</ENGLISH_TRANSLATION>
```

```
Translate conlang prompt

Translate the following text from "language" to English. language is
a constructed language spoken by species on the planet planet.

<TEXT_TO_TRANSLATE>
source
</TEXT_TO_TRANSLATE>

To help in the translation, here is detailed information about the
culture and language language.

conculture

conlang

# Instructions

**Recall that the only text you are translating is the following,
based on the above description of language:**

<TEXT_TO_TRANSLATE>
source
</TEXT_TO_TRANSLATE>

Just output your translation (no commentary) in the following format:

<TRANSLATION>
(english translation)
</TRANSLATION>
```

Figure 8: Our templates for translating low-resource languages, used for Wikipedia (top) and conlangs (bottom).

| Language | Unique properties | Species / Planet | Conculture + Conlang |
|---|---|---|---|
| Hran'yu | Adults can invert their subjective time flow for brief intervals, and their speech pairs forward- and backward-time syllables so recipients can 'pre-hear' replies; grammar marks retrocausative commitments. | Kelith / Ulithra | 80,483 chars |
| Sidiku | Each adult splits into up to 64 mobile lithic shards that remain a single mind with zero latency; they converse via phase-locked microseisms where sentences are spatial chords rather than linear sequences. | Shard Choir / Qas'dur | 67,464 chars |
| Vekhar | Their photonic-crystal skin sustains coherent light-solitons that are detachable 'thoughts' traded between bodies; syntax is encoded in soliton phase, polarization, and delay. | Rhyzont / Luminis-F | 72,316 chars |
| Talhi | They can switch the molecular chirality of their entire biochemistry on command; messages are broadcast as traversing chirality fields that pass through matter and support duplex conversations without crosstalk. | Naruun / Athiri-4 | 78,579 chars |
| Quol | They maintain swarms of external organelles ('exoviscera') orbiting the body; utterances are geometric rearrangements of these organelles into transient constellations that convey grammar and intent. | Cirriven / Xyr-Polis | 87,926 chars |
| Serren | They periodically flatten into ultrathin superconductive films and speak by imprinting non-decaying topological current knots; writing is stored as persistent braided currents in crystal veins. | Aelikat / Pethra | 78,369 chars |
| Avaru | They grow quantized gravitational vacuoles that emit controlled microgravity chirps; discourse uses curvature signatures and allows temporary mind-merging by braiding vacuoles into shared wells. | Quens / Otholome | 78,659 chars |
| Hushuun | They host two interleaved biochemistries oscillating 2 milliseconds out of phase; communication exploits interference between phases, enabling self-addressed messages from immediate future selves and lossless communal recall. | Merethi / Nidon-Beta | 76,890 chars |
| Odrial | Individuals are federations of interchangeable limbs with transferable ownership; pronouns index limb provenance, and clauses can change author mid-utterance as limbs vote and reassign. | Ythre / Serqa | 72,894 chars |
| Tuliq | They sense and emit neutrino flavor oscillations via specialized 'flavor-glands,' conversing through hundreds of kilometers of rock; grammar rides controlled baseline shifts, enabling absolute timestamping and time-locked messages. | Sookh / Helikon-7 | 74,382 chars |

Table 1: Summary of the ten conlangs generated using GPT-5. Each conlang has 10 sources with reference translations as well.

**Conlang ideation prompt**

```
We are creating a diverse set of conlangs about an alien species.
First, we are choosing the:
* Name of the planet
* Name of the species
* Name of the language
* The script, which should not be very common like the Latin script.
It can be something rarer like the Telugu script.
* Unexpected and unique property of the species that is not known in
any Earth lifeform

Output a JSON list of number_of_conlangs such objects, each with the
following keys:

Output a JSON object with the following keys:

{
    "planet": "Name of the planet",
    "species": "Name of the species",
    "language": "Name of the language",
    "script": "Name of the script",
    "property": "Unexpected and unique properties of the species and
    communication that is not known in any Earth lifeform (including
    humans)"
}
```

**Conculture generation prompt**

```
Create a vivid, detailed, and imaginative "conculture" (constructed
culture) for the species alien species who inhabit the planet planet.
They are unique in that the following sense: property.
The conculture should describe the planet and species in enough
detail to write a novel about one of the aliens.
The conculture should include detailed descriptions (e.g., at least
800 words each) of five practices (e.g., games, rituals, social
norms, etc.).
Their language, language, is written in the script script, but do not
detail that here. That will be defined later.
```

Figure 9: Our templates for ideating 10 different conlangs together, along with an elaborate "conculture" for each.

---

**Conlang language-definition prompt**

```
Create a "conlang" (constructed language) called language for the
species aliens described below. It will be written in the script
script but the language itself will not resemble any human language.

<CONCULTURE>
conculture
</CONCULTURE>

The language conlang should be unique in at least one unexpected way
that differs from any known existing language.
As background, describe the fascinating communication patterns of the
species in detail. Their communication must be entirely different
from Earth species---so much so that a naive translation into English
would be not be comprehensible without this background.
The description should be long and detailed, especially the grammar
and lexicon. The structure of conversations, meetings, and common
topics should be detailed. If there are multiple dialects, just
define and describe one.
```

Figure 10: Our template for creating a conlang based on a detailed description of a alien species and their conculture.

Finally, we create ten parallel texts based on the prompt in Figure 11. These were divided into sentence segments, unlike the Wikipedia experiment which is based on paragraphs.

Full specifications of all conlangs along with sample translations will be made available upon publication.

### D.2 FURTHER DETAILS FOR CONLANG EXPERIMENTS

As in the low-resource Wikipedia language experiment, we also tested the raw shuffle test without translation, by running the test on the sentences of the English reference translations. As seen in Figure 12, accuracies were significantly lower than in the case of Wikipedia articles. This may be because of the unfamiliar, alien nature of the languages. Nonetheless, GPT-5 achieved a remarkable 94% accuracy. Also note that the swap test has been found to be more challenging on longer-length segments, like paragraphs.

## E HALLUCINATIONS IN WHOLE-ARTICLE TRANSLATION

Note that transformer-based LMs are well-known to hallucinate (Huang et al., 2025) in translation and other contexts. To illustrate this risk in our data, we took the Wikipedia articles (which as mentioned had been truncated to six paragraphs) and translated them using the same prompt of Figure 8. We then used an RFQE prompt similar to the standard one in Figure 4 to evaluate it on its own merits. We find that one LM may score higher than another in the standard RFQE due to hallucinations. One case is GPT-4o-mini, which scored higher than GPT-5 in terms of this RFQE on whole-document translations. More than 94% of the translations scored 90 or higher out of 100 for both models. Among these, however, Figure 13 shows that many GPT-4o-mini generations score very low when scored using our baseline reference-based MTQE of Figure 4.

### E.1 DETAILED WIKIPEDIA HALLUCINATION EXAMPLE

We now expand on the example from Figure 2. We provide: (a) three document-level translations of the article (truncated to the first six paragraphs), (b) the first six paragraphs of the English version

```
Conlang parallel-text generation prompt

Create 10 texts in the alien conlang language spoken by species,
described below. The texts should be of varying lenths, with the
shortest one being 6 sentences and the longest one being 20
sentences. Each text should have an English translation.
At least 5 of the texts should rely on detailed descriptions of the
species and practices/peculiarities in the <CONCULTURE> section
below. It is fine if the texts use vocabulary not defined in the
conlang below, just add it to the additional_vocabulary section.

<CONCULTURE>
conculture
</CONCULTURE>

<CONLANG>
conlang
</CONLANG>

Your output should be in JSON with the following structure:

{
    "texts": [
            {
                "language": [list of strings for sentences],
                "English": [list of strings for sentence translations]
            }
        ... # 9 more texts
    ]
    "additional_vocabulary": # long string with describing the
    additional vocabulary needed to understand the texts, if not
    present in the conlang above
}
```

Figure 11: Template for creating ten texts in a given conlang, based on conculture, along with reference English translations.

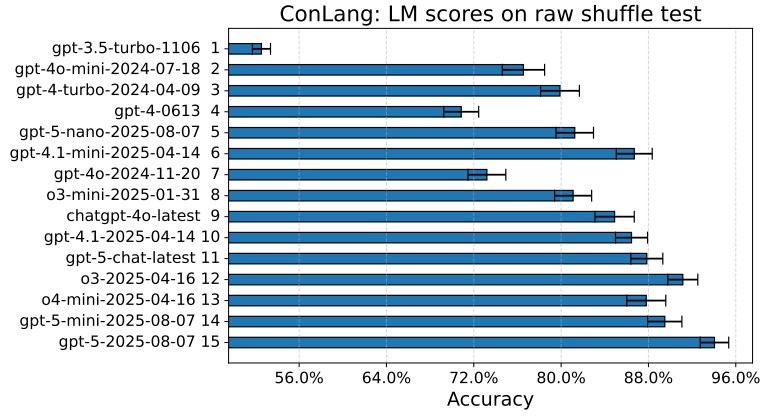

Figure 12: Judging the judge for the ten artificial conlangs. The LM is tested for its accuracy at determining the original order of the reference sentences. Without translation, accuracy at distinguishing the original paragraph order from a permutation on the English references. The fifteen LM's evaluated ranged from GPT-3.5-turbo (1) to GPT-5 (15).

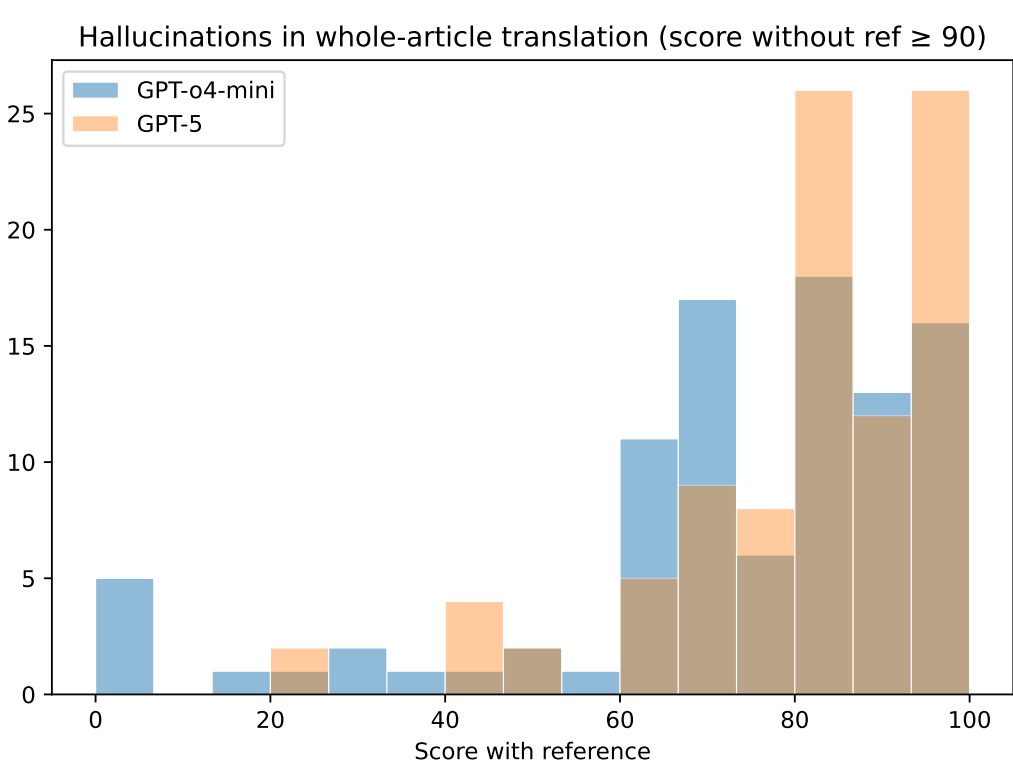

Figure 13: Baseline reference-based scores of GPT-5 and GPT-4o-mini translations *with reference*, among "plausible" translations as judged by having a RFQE score of $\geq 90$. Many GPT-4o-mini translations appear to be likely hallucinations, scoring 0 out of 100 when compared to the reference.

of the Wikipedia article on the *Patna–Digha_Ghat_line*.[4], (c) three paragraph-by-paragraph translations.

The source in Santali:

ᱪᱟᱝᱱᱮᱫ-ᱯᱟᱜᱟᱝ ᱜᱮᱜᱟᱝ ᱯᱟᱜᱟᱭᱮ ᱯᱚ ᱪᱟᱝᱱᱮᱫ ᱱᱚ'ᱵᱟᱜᱟᱱ ᱢᱟ ᱯᱟᱜᱟᱝ ᱜᱮᱜᱟᱝ ᱥᱮᱯᱟᱝᱮ ᱠᱱᱮᱠᱟᱮ ᱯᱚ ᱠᱟᱺᱚᱚᱟᱜᱟᱝ, ᱠᱚᱣᱚᱝ ᱪᱟᱝᱱᱮᱝ ᱥᱮ ᱩᱭᱯ ᱥᱮᱯᱟᱝᱮ ᱯᱟᱜᱟᱝᱮ ᱚᱝᱣᱮ'ᱶ ᱵᱟᱢᱮᱝ ᱹ ᱮᱚᱩᱢ ᱯᱚ ᱹᱮᱮᱮ ᱠᱟᱯ ᱥᱮ ᱚᱥᱮᱱᱥᱮ ᱵᱚ ᱚᱥᱮᱢᱩ ᱯᱥᱯ ᱚᱝᱣᱮ'ᱶ. ᱢᱟ ᱮᱚᱩᱢ ᱯᱚ ᱚᱝᱯᱟ ᱚᱝᱯᱟ ᱚᱥ ᱺᱟᱯᱟᱢᱩ ᱯᱥᱮᱝ ᱹ ᱮᱚᱩᱢ ᱥᱮᱯᱟᱝᱮ...

The translation of the first six paragraphs by GPT-3.5-turbo (LM 1):

*The Wild Animals and Their Association*

*By the tender age of four or five, when our bodies are neither weak nor strong, the adivasi children probably learned to live with bravery and dignity without even feeling it. Indeed, we can suspect they gain strength from the innocent aspects of wild animals and perhaps have inherited the characteristics of those animals. They perhaps believe that the length of time they spend on the ground, without sophistication, is as strong as the tenderness of a wild boar which can readily destroy the obstacles, when rarely needed. Such a forest dweller will not tolerate wrong even in his dreams; this shows that great men have sprung up among them who are like the great animals in the forest involving efforts and with endurance. They can attain freedom from fear, either easily or with strain, as easily as these wild animals do.*

*According to the year 1900 or 1901, the folks had gained immense courage and strength over time. However, they are living a life of hardship, with poverty. At this time, these adivasi people realize the wonders of the wild forest and learn to respect and understand the life of the wild animals which keeps their households well.*

*A Healthy Morning and Evening with Domestic and Wild Animals*

*The adivasi race is able to lead an active life by spending half their day in the hilly terrains and the other half with the domesticated animals. Due to the strength and resistance built by spending time in the forest and the hills, the adivasi people only begin to feel tired at around 7 or 8 p.m., and the domestic and wild animals keep everything harmonious with the bond they share. For fear of extinction of themselves and their animals, the Santhals had gradually preserved and carefully tried to protect both forms of life. They did so by dividing their time, half with wild animals and the rest with the other domestic animals, obviously with a responsible heart. As a result, their names were eternally recorded in the history, culture, and the love they had for their domesticated animals through a mutual bond and careful nurturing.*

*The midnight of the year 1908 (1912 AD).*

*However, when they listened carefully and found they had saved and protected those animals in time, they probably felt a sense of happiness on the day when they were able to prepare and consequently preserve the well-being of human and animal alike.*

*With Peace and Harmony, they try to By-Life with Animals*

The translation of the first six paragraphs by GPT-4 (LM 4):

*The Sarna-Liturgical Kanthi (rosary) of the Sarna faith, the independent religious system of the indigenous people, and the sacred text of Sarna-Liturgical came to light in the light of detailed research, conversation, and meticulous efforts. According to the 2011 census, Sarna is a believer of 46,73,848 people in the country, and he is recognized as a primalmul (adivasi) by the constitution of the country. In the context, 14,73,931-48,64,238 out of them and 40.14% of the total population of the state are Sarna. The historic Sarna-Liturgical tradition is flourishing in most of the local languages (like Kurukh, Kudukh, Nagpuri, Bengali etc.), the independent and immeasurable religion of the Sarna people, being found in different Socio-cultural and linguistic background of various tribes in the state it is practicing.*

*Since 1921, the unique religious traditions of the Adivasi-Sarna and the Sarna-Dharam traditions have been covered by karma and festivals and their complexities have not been exposed, thus cre-*

---

[4] https://en.wikipedia.org/w/index.php?title=PatnaDigha_Ghat_line&oldid=1220622116

*ating confusion in the minds of outsiders and even among many of our own. Therefore, since 1941, Sarna's distinctive religious rituals, festivals, rituals, rituals, etc., are largely unwritten, conventional, and have long been conveyed by mere faith and practice. It is late but it is time to write it down.*

*Threats*
*Sarna-Faithful Kanthi or Astronomical Kanthi and its traditions have been under threat since the 1891-1950 period has brought about decolonization, Christianity and rapid change in the state of economic, social and cultural life. These threats have grown significantly in the post-globalization period of 1991-2020 and have taken many forms and serious turns. The important resources of Sarnaism like Gotul system, the indigenous education system, the wealth and purity of nature, the centuries-old forests have been destroyed or are on the verge of being destroyed. In the 2011 census, the number of Sarna believers has reduced from 46,73,848 to 46.74 percent and from a social point of view, the decisive population is 25,17,958, so the Sarna scriptures have been taken to raise serious questions and challenging situations. Protection*
*The Sarna-Liturgical Kanthi has been strictly adhered to and is written to retain its traditional and independent form, which is the main reason why rituals have been performed and emphasized for centuries. Even into the twenty-first century, where the developed world is moving towards the smart world and digital civilization, the Sarna religion of the original inhabitants is holding a lot of attention. The term "Non-Scripture" applies to Sarna ethics and moral education (such as making man wise, useful to society, aware of right-wrong / wellness, etc.), all ingrained in the minds of the community by dozens of stories, various legends etc. The Sarna religious Kanthi has been adopted and propagated for the preservation of the invaluable and diverse elements of Sarna Economy, the knowledge and practice of man's physical, psychological, and social health. The unique religious rituals of the Sarna-Liturgical Kanthi have to be strictly adhered to and ensured to retain its traditional and independent form, which has been the basis of its original form since the 9th century (ie AD 850) till date. The total faithful population of Jharkhand, which is 9.9 percent (47.1 million), is a total believer.*

*Necessity*
*The human race, which is 46,73,848 souls in the world, has spent centuries in the background of indigenous Sarna-Dharam Kanthi. The parrot does not have an independent, universal, common, and scriptural religious volume. The humiliation of Parrot before the scholars and intellectuals is a definite conclusion. The period from 1991 to 1993, considering the Sarna religious Kanthi from the point of view of the educated society, the absence of an independent, universal, common and scriptural religious volume, is a necessity. Therefore, post-education, self-development, globalization management and community development of the educated community, requires the Sarna religious Kanthi to be scripted as an independent, universal, common, and scriptural religious volume.*

*Necessities*
*The basic needs, the need for a universal, common, and scriptural volume of the Sarna religious Kanthi, considering the view of educated society, it is indeed necessary. Social awareness of the Sarna Religious Kanthi, a population of 14,73,931-48,64,238, the total population of the country (14,37,723), is discussed in proportion to the need for an independent, universal, and scriptural religions volume for the Saunas religious Kanthi. The restoration of the Sarna religious tradition became necessary.*

*Procedures*
*The Sarna-Liturgical Kanthi, the household ritual of Sarna religious liturgical, has been proposed in several religious rituals (household worship, weekly, monthly, annual festivals, death anniversary etc), with the people in the heart of Sarna-Dharma folk practices in their heart of hearts.*

Paragraph-by-paragraph translation by GPT-5 (LM 15):

*Patna–Digha Ghat line is a railway line between Patna Junction and Digha Ghat railway stations in the city of Patna, Bihar. It was built by the British in 1862, and later trains were operated on it. In 1962–63 and again in 2004 the Railway Ministry, and Lalu Prasad Yadav, took steps regarding this line. However, regular passenger train services could not be run on this route; instead, it was used to carry sand and stone. Because of encroachments, it was difficult to run trains on the line. In August 2018, the railway line and its associated structures were handed over to the Bihar government to help reduce urban traffic congestion; consequently, the rails along the entire*

*line were completely removed. The project was slated to be fully completed in 2021. In addition, for building the Block–Digha Road, the Bihar government took up work on Hardinge Road as a 7-kilometre corridor, and 1.9 hectares (4.8 acres) of land from the Patna–Digha Ghat line's route was also taken.*

*History*

*Under British rule, in 1862 the Patna Junction–Digha Ghat rail track was built. Because Digha had factories and for the movement of freight and goods, the railway line was constructed. In 1962–63 the Railway Minister took up renewal of this line, and passenger train service started in 1974. In 1990 goods trains also began operating on the line. However, from July 2004 the track was scarcely used, although Railway Minister Lalu Prasad Yadav had restarted slow passenger train service on it.*

*Route*

*The Patna–Digha Ghat railway line was 6.2 kilometres (3.9 miles) long. It ran between Patna Junction railway station and Digha Ghat railway station, with several intermediate halts (Rajiv Nagar, Shivpuri, Punaichak, Bailey Road and Secretariat), and there were many at-grade road crossings along the route. The line ran almost entirely alongside Bailey Road, and because of this and the numerous crossings, running trains on the line was difficult.*

*Notes*

*In earlier years, the railway line was used for transporting products and raw materials to and from the Bata factory at Digha, which made movement of goods easier. Furthermore, the track was used by the municipal corporation to carry garbage to the dumping ground at Digha.*

The first six paragraphs of "Patna–Digha_Ghat_line" in English Wikipedia:

*Patna–Digha Ghat line was a railway line connecting Patna Junction and Digha Ghat railway station in the Indian city of Patna. It was constructed by the British in 1862, and functioned intermittently. The railway line was revived in 1962–63 and 2004 by the then Indian Railway Ministers, Jagjivan Ram and Lalu Prasad Yadav, respectively. Passenger train operations on the route were economically unviable due to low ridership. However, trains continued to run on the railway line in order to prevent encroachment.*

*In August 2018, the railway line along with the surrounding land was transferred to the Bihar state government, which plans to convert it into a four- to six-lane roadway to ease traffic congestion in the city. The project is scheduled to be completed by mid-2021. Indian Railways got around of land on Hardinge Road from Bihar government in lieu of the 7 km long Patna–Digha Ghat line land it offered to the latter for the construction of R-Block-Digha Road.*

*History*

*The Patna Junction–Digha Ghat rail track was laid in 1862 during the British Raj. The railway line was constructed to ease the transportation of leather items and footwear products to and from the Bata factory situated at Digha.*

*The railway line was revived in 196263 by Jagjivan Ram, the then railway minister of India, citing the historical importance of the line. The passenger train service continued until 1974, with goods trains running on the railway line until the early 1990s. Subsequently, the track remained non–functional until July 2004 when the Minister of Railways, Lalu Prasad Yadav, restarted the shuttle-train service along the line.*

*Route*

### E.2 SEGMENT-LEVEL TRANSLATIONS

We now present the segment-level translations.

The translation of the first six paragraphs by GPT-3.5-turbo (LM 1):

1. *The consequences of forest fire. Now a days, we are facing new challenges due to climate change. Forest fire is increasing day by day, so we are losing many herbal plants and trees. If this continues, we will face another disaster that our future generation will not forgive*

*us. We need to work together to prevent forest fire and protect our environment, so that we can leave a beautiful earth for our future generation.*

2. *In the year 1894, the British government started to rule over our ancestors, therefore, the Great Santal Hul broke out as a means of revolutionary movement against the British. As a result, the Santals lost many lives and properties, and even now are suffering from that loss. Since the Great Santal Hul, the Santals have been continuing their struggle for rights and identity in various ways today, aiming at safeguarding their own existence and culture. The Santals are a hereditarily operational people, but to incorporate with modern civilization, they have been embracing modern education. There are evidences that they are able to achieve good results through this education. Their interest in education is growing day by day.*

3. *The world is characterized by inequality, injustice, and exploitation. Many people in our society are still struggling for their rights and justice. We need to work together to create a world where everyone can live with dignity and respect.*

4. *In the year 1952-53, our ancestors originated the concept of governance through traditional customs and a council of elders. They also introduced a system of selecting their leaders based on merit and integrity. In the year 1974, the movement for the recognition and rights of the indigenous people came into being. They struggled for autonomy and identity, and ultimately achieved their objectives.*

5. *The crow and the water pot. Once upon a time, there was a crow. It was very thirsty. It was searching for water. At last, it found water in a pot. But the water in the pot was very low. So, the crow thought of a plan. It picked up stones and put them into the pot and the water level went up. Then the crow drank water and flew away.*

6. *A tiger and a man*
   *Once a man, while going through the forest, saw a tiger cub. Then the tiger cub grew up and became a ferocious tiger in the jungle. One day, the man went to cut wood on the mountain. As soon as the tiger saw the man, it ran to eat him. But the man thought of something and lied to the tiger. The man said, "My elder brother is coming here to cut wood." Thinking of the man's words, the tiger thought that it would not eat the man.*

Paragraph-by-paragraph translation by GPT-4 (LM 4):

1. *"Serma-Ravana was a rich man from the city of Lanka, and Ravana was the king of a huge state who lived comfortably. Suddenly, a severe famine broke out in Lanka, and food and water became extremely scarce. In 1892, about two million people died of starvation, and the dead bodies were rotting all around. In the streets, people were dying of hunger, and the smell of the carcasses was unbearable. Between 1896 and 1906, approximately 20 million people died from starvation and the situation became so critical that the British government had to declare a state of emergency. Many people left their homes in search of food and migrated to other places. To tackle the famine, the British Government introduced several relief programs like opening free kitchens, granting loans and resources for food. Despite the measures taken to counter the famine, the death toll continued to rise, and people lost their trust in the government. This led to the publication of a report about the mismanagement done by the government in handling the famine situation, which deeply affected the people of Lanka."*

2. *On August 30, the whole world celebrates International Day of the Disappeared. This day is dedicated to the issue of missing people, especially in situations of violence or armed conflict. It is a day that raises awareness about the fate of individuals who have suddenly disappeared and the deep suffering that their families and friends face. The International Day of the Disappeared was first recognized globally in 2011 and since then, it has been an annual event that commemorates and brings attention to the horrific act of enforced disappearance. Through this day, the spotlight is put on the victims who have disappeared in different parts of the world and the importance of addressing this issue.*

3. *Dreams*
   *Since the beginning they were with us. Around 2690 BC people even started analyzing-needing them in reality. The Chinese also considered dreams to be a way to communicate*

*with the dead, as a way to get divine information. Egyptians also considered dreams to be answers to future riddles. This means we can learn from our dreams.*

4. *In 1939-40, Sir John Maffey's proposal for constitutional reforms in India was accepted by the British Government. Under this, in the state of Bihar, the Santali Pargana district was merged into the Manbvum district and made a part of Bihar. But in 1947, the majority of the people of Bihar rejected the trial of the people, which meant that they wanted to live as a separate district, not part of Bihar. In 1950, the Santal Pargana district was separated from Bihar and made a part of West Bengal.*

5. *February*
*Sarna-Religious flag raising festival of Sarna believers. The tradition of raising the jari (56 kg) sarna-sacrificial flag (56 in number) at the rate of one each on all the four sides of the sarna-prayer ground was started by Sadhu Ramchandra Murmu of Dumka, which is still being followed. Some rituals performed during this flag raising festival involve animal sacrifices (e.g., goats, sheep, pigeons, quails, and turtles) and sarna-worshippers believe that on this day, it is forbidden to touch anyone or anything when the sarna prayer ground is cleansed or when the Sarna-prayer ground's flag is being replaced. Those who disobey this rule will be turned into turtles.*

6. *He jumped*
*Thinking about the end of the month, the poor guy is on his heels all day, and after a hard day, he thinks about the next day before he goes to bed. The thought of the next day's income and expenses made him scream and the peace of his heart is disturbed. A sigh, the poor fellow has to think about the beginning of the month day and night, then if his income is lower than his expenditure, he has to think about borrowing,*

Paragraph-by-paragraph translation by GPT-5 (LM 15):

1. *The Patna–Digha Ghat line is a metre-gauge railway line between Patna Junction and Digha Ghat railway station, located in the city of Patna. It was built in 1862 by the British and ran for a long time. In 1962–63 and again in 2004, Railway Ministers Jagjivan Ram and Lalu Prasad Yadav, respectively, took decisions regarding this line. In later years, converting it to broad gauge made running passenger trains impossible, so services were stopped. At present, due to encroachments, trains cannot run on the line.*

2. *In August 2018, the Bihar state government announced that the land of the railway line and its ancillary structures would be used to convert the railway alignment into a road, creating a new traffic corridor to reduce the frequent congestion seen in the city. This project will be completed in 2021 according to the estimated cost. To build the Block–Digha Road on the railway alignment, the Bihar government acquired, along Harding Road, 1.9 hectares (4.8 acres) of land from the 7-kilometre-long Patna–Digha Ghat railway line.*

3. *Name*
*Under the British Raj, in October 1862, construction of the Patna Junction–Digha Ghat rail track began. At Digha, to transport factory and market goods more easily, a new railway line was built.*

4. *In 1962–63, Railway Minister Jagjivan Ram, in view of the Naga movement along the border line, had two new roads laid. Passenger train service began in 1974, and in 1990 goods trains started operating on the railway line. Later, in July 2005, work on a new track began. When Railway Minister Lalu Prasad Yadav was in office, a slow/local train service was introduced on this line.*

5. *Overview*
*The Patna–Digha Ghat railway line was 6.2 kilometres (3.9 miles) long. It ran between Patna Junction railway station and Digha Ghat railway station, with several small halts on the route—Rajiv Nagar, Shivpuri, Punaichak, Bailey Road and Secretariat. Later the tracks were removed and the right-of-way was converted into a road as the Digha Corridor. The work of dismantling the railway track along this line has been completed, and the railway line here has been closed.*

6. *Problems*
*In our locality, the railway line has not been in use for a long time; railway materials have*

> *been piled up near a roadside workshop, and because of this the public/passers-by face hardship, with things scattered everywhere and the place looking very untidy. Furthermore, the railway track has had no maintenance for a long time, and it is now being used by the Corporation as a warehouse for storage.*

As one can see, not only do the GPT-4 and GPT-3.5 paragraph-by-paragraph translations fail the shuffle test, but they change topic repeatedly. However, staying on topic is not a sufficient validation for translation. Thus, while one may incorporate a penalty for changing topic, merely producing coherent text that stays on topic may not be considered a validation.

# F  PROOFS FOR SECTION 2

*Proof of Theorem 1.* The proof is a simple application of Hoeffding's inequality, following the proof of the standard Occam bound of binary classification (Kearns & Vazirani, 1994). Let $\eta := \sqrt{\frac{\ln(|\mathcal{F}|/\delta)}{2m}}$. Note that the term on the RHS of Eq. (3) is $2\eta$. Let $f^\star$ be any minimizer of $\ell_\mathcal{D}(f)$, so $\ell_\mathcal{D}(f^\star) = \text{opt} := \min_{f \in \mathcal{F}} \ell_\mathcal{D}(f)$. Let $\hat{\ell}$ denote the empirical loss (Eq. (2)). We claim that Eq. (3) holds as long as,

$$\hat{\ell}(f^\star) \leq \ell_\mathcal{D}(f^\star) + \eta, \text{ and}$$
$$\hat{\ell}(f) \geq \ell_\mathcal{D}(f) - \eta \ \text{ for all } f \neq f^\star$$

because in this case all "bad" $f$ with $\ell_\mathcal{D}(f) \geq \text{opt} + 2\eta$ have $\hat{\ell}(f) > \text{opt} + \eta$, and $f^\star$ is a non-bad option for $\hat{f}$. By Hoeffding's inequality, each of these inequalities fails for a given $f \in \mathcal{F}$ (either $f = f^\star$ or $f \neq f^\star$) with probability $\leq \delta/|\mathcal{F}|$. Thus the probability that any "bad" $f$ is in $\hat{\mathcal{F}}$ is at most $\delta$ (union bound). □

*Proof of Corollary 2.* Take $m := n/\varepsilon c$ training observations. By definition of $\varepsilon$, this costs at most a $1/c$-fraction of $n$ interactive experiments. Invoking Theorem 1 with $\delta = 0.01$, then with probability at least 0.99,

$$\ell_\mathcal{D}(\hat{f}_n) \leq \text{opt}_n + \sqrt{\frac{2\varepsilon c \ln(100 \cdot |\mathcal{F}_n|)}{n}}.$$

The proof is completed using the fact that $\ln(100) < 5$ and $2\ln|\mathcal{F}_n| \leq 3\log_2|\mathcal{F}_n|$. □

