# OpenReview forum: "On Non-interactive Evaluation of Animal Communication Translators"
_ICLR.cc/2026/Conference — Submitted to ICLR 2026_

### Official Review · Reviewer_FqGT · 2025-10-26

**Soundness:** 2
**Presentation:** 3
**Contribution:** 3
**Rating:** 8
**Confidence:** 3

**Summary:**

The paper proposes and evaluates a method on how to rate the translation quality of translations which don’t contain any reference.
The proposed method is to split the given source into segments (using a LLM) and translate each segment independently (using a LLM). Afterwards a LLM is used to judge if the ordering of the target sequences makes sense.

It’s a inneressing approach to solve an unusual problem. The main weakness seems to be that it’s overly relying on wikipedia and models from the same LLM company.

**Strengths:**

The method allows to give some insights of whether the translation makes any sense or not when no reference is available.

For animal communication it allows one to get some impression if the method works or not without requiring additional communication, observing the animals is enough.

They confirmed that the approach works on low resource languages and on constructed languages.

**Weaknesses:**

I think the results shown in Figure 5 are most likely exaggerated. While it’s very plausible that later models simply perform better because they are strong, it is also very likely that later models saw the given wikipedia articles in its latest form during training. These models would know the correct ordering which in turn would inflate their results.

Since the parallel text were extracted from wikipedia articles in different languages there is a high likelihood that the first paragraphs are not really translations of each other. The authors acknowledge this, but not having proper translations still adds a lot of noise into the evaluation.

We can assume that all LLMs saw the latest wikipedia version available at training time. It’s not unlikely that they learned the correct paragraph ordering across languages. Also since the approach to treat the first paragraphs as parallel data seems to work, we can assume that the ordering is similar across languages meaning the LLMs saw the right ordering during training.

Every step in this work involves LLMs from OpenAI. This increases the likelihood that the LLM actually knows the correct sentence since it was created by a similarly trained LLM, again inflating the results. It would be good to see the results of at least one other LLM as well. I don’t expect a different result (thanks to wikipedia), but it would be more valuable information than having everything from related LLMs.

The constructed languages were created using LLMs. Given that LLMs are trained on human languages, I’m not really convinced that these languages are really so different to human languages as claimed. For the given purpose I think the described method should work well enough.

**Questions:**

End of Line 255: I think you forgot the “=0” and wanted to write “p(T, T’) = 0 indicates T is more plausible than T’”.

Was the cut off date June 1st, 2024 only checked on the non English side of the wikipedia article, or also on the English side?

---

> ### Author Response · Authors · 2025-11-22
>
> Thank you for the encouraging review, and for your constructive questions and criticism. We fixed the typo you correctly pointed out (l. 255), and address the concerns next.
>
> > We can assume that all LLMs saw the latest wikipedia version available at training time… While it’s very plausible that later models simply perform better because they are strong, it is also very likely that later models saw the given wikipedia articles in its latest form during training
>
> We reiterate that only Wikipedia articles written after June 1st 2024 were used whereas all but the GPT-5 models have a knowledge cutoff date earlier than that. To rigorously address your concern of data contamination, we compared the hypothesis "Training Date drives ordering accuracy" against our hypothesis ("Model Scale drives ordering accuracy"). The analysis conclusively refutes the concern: a Mann-Whitney U test shows no significant difference between post-cutoff and pre-cutoff models ($p=0.21$), making the "Post-Cutoff" feature statistically indistinguishable from noise. In contrast, Model Scale is a far stronger predictor of rank ($p=0.077$, Spearman’s $\rho \approx -0.47$). As a concrete example, note that GPT-5-nano (which may have "seen" some articles) is significantly outperformed by "blind" but larger models (o3,o4-mini). We append full details of this analysis, and will add it to the paper. Thank you for this important question which prompted us to strengthen this aspect of our paper.
>
> > Every step in this work involves LLMs from OpenAI. This increases the likelihood that the LLM actually knows the correct sentence since it was created by a similarly trained LLM, again inflating the results.
>
> Our experimental design relies on context isolation to prevent the data leakage or "telepathy" you describe. As detailed in Appendix D.1, the conlangs are generated in an involved pipeline and defined entirely within a given context window during the translation task. Because the "correct" translation rules exist only in the prompt (not the training set) the evaluator must rely on in-context learning rather than memorized patterns shared across the model family. This concern is also quantitatively refuted by Figure 12: there are statistically significant gaps between various models.
>
> > Given that LLMs are trained on human languages, I’m not really convinced that these languages are really so different to human languages as claimed. For the given purpose I think the described method should work well enough.
>
> We agree with the reviewer. A degree of human bias is inevitable when generating languages that must, by definition, have English reference translations to allow for validation of the methodology. We appreciate the reviewer’s assessment that the method works well enough for the intended purpose; together with the Wikipedia-based experiments, we believe this suffices to support the central claim of the paper.
>
> > Was the cut off date June 1st, 2024 only checked on the non English side of the wikipedia article, or also on the English side?
>
> The cutoff date was checked only on the source side. However, this distinction does not jeopardize the methodology: Because the source text is post-cutoff (and thus unseen), the model cannot rely on a memorized mapping between the input and the target. Even if the model theoretically "knows" the older English article, it has never encountered the specific source tokens required to trigger that retrieval. Consequently, it must rely on actual translation capabilities to map the novel source text to English.

---

> > ### Author Response · Authors · 2025-11-22
> > **Rebuttal Appendix: Statistical Analysis of data contamination concern**
> >
> > We tested two competing hypotheses to explain the ranking data:
> >  1. Hypothesis A (Contamination): "Post-Cutoff" models outperform "Pre-Cutoff" models.
> >    - Test: Mann-Whitney U Test (Two-sided).
> >    - Result: p=0.21. Null hypothesis is not rejected.
> >
> > 2. Hypothesis B (Capabilities): Using parameter count as a proxy for capability (acknowledging that later architectures tend to have better capabilities given the same number of parameters), larger models perform better than smaller ones.
> >  - Test: Spearman’s Rank Correlation (rho).
> >  - Result: p=0.077 with a correlation of rho approx -0.47.
> >  - Conclusion: While recent architectural shifts (e.g., o-series efficiency) introduce variance, Model Scale remains a far stronger predictor of rank than Training Date (p approx 0.08 vs p approx 0.21).
> >
> > Data Summary
> > Model parameter counts are not publicly available, so we gathered publicly-available estimates and used midpoints as the number of parameters. "Rank Tier" refers to rank accordingto  the score depicted in Figure5.
> >
> > | Model | Rank Tier | Est. Params | Group |
> > | :--- | :--- | :--- | :--- |
> > | GPT-5 | 1 | ~3.5T | Post-Cutoff |
> > | GPT-5-chat | 2 | ~3.5T | Post-Cutoff |
> > | GPT-5-mini | 2 | ~500B | Post-Cutoff |
> > | o3 | 2 | ~250B | Pre-Cutoff |
> > | o4-mini | 2 | ~150B | Pre-Cutoff |
> > | GPT-4o | 3 | ~200B | Pre-Cutoff |
> > | GPT-5-nano | 4 | ~11.5B | Post-Cutoff |
> > | GPT-4 | 4 | ~1.7T | Pre-Cutoff |
> > | GPT-3.5 | 7 | ~175B | Pre-Cutoff |

---

### Official Review · Reviewer_guKa · 2025-10-31

**Soundness:** 3
**Presentation:** 2
**Contribution:** 2
**Rating:** 4
**Confidence:** 3

**Summary:**

This submission derives some theoretical results on learning based on interactions vs. (less invasive and easier) observations. This is provided in the context of reference-free evaluation of low-resource or no-resource translation, with the particular use case of the translation of animal communication. Experiments show that the proposed shuffling-based metric is potentially useful to evaluate translation, without requiring costly or impossible to acquire references.

**Strengths:**

The core problem of evaluating "translations" in the context where references are not available is highly relevant. The application of evaluating animal communication is interesting and exciting. The proposed technique, relying on comparing the coherence of a translation with a shuffled version of itself, is novel (in this context). Experiments on both low-resource languages and constructed languages show that the proposed shuffle test yields significant correlation with a reference-based evaluation, which is promising.

**Weaknesses:**

A large part of the paper is occupied by a theoretical analysis. The theory it presents derives from well-established learning theory results. More importantly, the setup of interactive vs. observational learning is very loosely connected to the actual, highly interesting application to animal communication. In fact, there is essentially no reference to the results in Section 2 in the rest of the paper. A much tighter and clearer connection between that theory and the shuffle test would be highly appreciated. Otherwise, more methodological and experimental details on ShuffleEval and its evaluation (now exiled in the appendix) would be welcome.

Experiments are limited to about 100 test examples in each language/conlang. This may reflect operational constraints of animal communications studies, but is fairly low by machine translation standards. In that context, error bars or uncertainty evaluation would greatly help qualify how variable actual results and assess confidence.

Although experiments involve many "translators", they are all flavours of OpenAI's GTP. In addition, all evaluation is done using GPT. MT metrics (esp. reference-free) is a lively field of research, it is surprising that none of these metrics was included as reference. Minimally, the use of GPT5/4 are references for coherence & MT quality could be manually validated on a sample of examples.

**Questions:**

Recent work on MT metrics suggest that novel LLM-based metrics have strength for high-performing (high-resource) languages, but struggle to estimate mid-to-low performance systems (such as typically the case for low-resource languages, and one would assume conlangs).

(l.177) Why is the empirical risk minimization infeasible? Do you mean because of practical (e.g. multiple minima) or theoretical reasons?

(l.201) How is the translator family growing with the number of interactive experiments? Would it not be fixed for a choice of parameterization?

(l.230-238) Paragraph is not super convincing as the argument relies on ad hoc parameter choices (c, eps).

(l.249) Why would translators be different for paragraph-level (f) and segment-level (\phi)? Esp. with LLMs one would expect that they are the same.

(l.366) Why is the date of June 1, 2024 chosen?

(l.343) "prior work has validated": a reference would be nice here. Presumably you mean the refs. in l.052.

(l.357) How is 99.9% estimated from 100 examples?

(l.481-483) Could you expand?

Typos:

l.091: Set 'F' is only introduced later (l.144)

l.140: the the

l.222: Is opt_n the same as {f}^*_n?

l.255: missing value before "indicates T"?

l.343: we highly -> were highly

l.352: missing 'of'?

l.354: while model the -> while the?

---

> ### Author Response · Authors · 2025-11-22
>
> We thank the reviewer for their careful reading of our paper, and for their many helpful suggestions both in terms of content and typos. Next, we address the concerns and questions raised in the review.
>
> > the setup of interactive vs. observational learning is very loosely connected to the actual, highly interesting application to animal communication
>
> The setup of interactive vs. observational learning is directly connected to hypothesis testing in animal communication, where the "gold standard" is playback experiments, often with detrimental impacts on the animals studied. We explain this explicit motivation multiple times in the paper (e.g. lines 94, 165, 215, 485). The results in Sections 2 and 3 are complementary: Section 2 argues from theoretical first principles that observational learning can suffice in high-error regimes, providing the general justification for Section 3, which demonstrates proofs of concept in actual experiments using real translators.
>
> >  error bars or uncertainty evaluation would greatly help qualify how variable actual results and assess confidence.
>
> We do report a $\pm0.04$ 95%-bootstrapped confidence interval for the correlation coefficient in the text (e.g. line 376). However, for the analyses depicted in Figures 6 and 7, we maintain that from a principled data visualization perspective, the means (points) are sound and sufficient. Since the analysis investigates the correlation between aggregate metrics, means _are_ the effective data points. Including error bars would visualize the variability across individual test examples, a tertiary property distinct from the correlation of the aggregates. This would add visual clutter and potentially mislead the reader into focusing on the statistical separability of individual models rather than the linear relationship between the metrics.
>
>
> > Although experiments involve many "translators", they are all flavours of OpenAI's GTP. In addition, all evaluation is done using GPT.
>
> All models were accessed using a single API (OpenAI) simplifying our experimental setup; however, we used fifteen distinct models ranging from "nano" to full-scale, released over several years. These represent vastly different parameter scales, architectures, and post-training methodologies. Furthermore, our experimental design specifically mitigates the risk that shared pre-training data across a model family would inflate results. As detailed in Appendix D.1 , the conlangs are generated in an involved pipeline and defined entirely within the prompt's context window during the translation task.
>
> >  MT metrics (esp. reference-free) is a lively field of research, it is surprising that none of these metrics was included as reference.
>
> Our validation is compared to "gold standard" reference translations (Wikipedia and Conlangs), which serves as the proxy for manual validation. That said, if there is a specific standard MT metric whose inclusion you believe would significantly strengthen the paper and would increase your score, we would attempt to implement and include it in the final revision.

---

> ### Author Response · Authors · 2025-11-22
> **Continued rebuttal: Addressing the Questions section**
>
> > Recent work on MT metrics suggest that novel LLM-based metrics have strength for high-performing (high-resource) languages, but struggle to estimate mid-to-low performance systems (such as typically the case for low-resource languages, and one would assume conlangs).
>
> We are unsure which specific recent works the reviewer is alluding to regarding this failure mode for the specific class of metrics we employ. If the reviewer can point to specific studies demonstrating this issue in this context, we are happy to address them.
>
> > Why is the empirical risk minimization infeasible? Do you mean because of practical (e.g. multiple minima) or theoretical reasons?
>
> In practice, computing the argmin for the empirical risk minimizer requires enumerating over all hypotheses, which is exponential in the number of parameters.
>
> > How is the translator family growing with the number of interactive experiments?
>
> The "Whalebreak" model assumes that interactive experiments are maximally efficient, with each experiment resolving $b$ bits of uncertainty. For this perfect efficiency to persist over $n$ experiments, the size of the hypothesis family must effectively grow $n$. By demonstrating that observational learning remains competitive even against this hypothetically optimal interactive learner, we establish its value without being constrained by the saturation limits of a fixed architecture.
>
> > Paragraph is not super convincing as the argument relies on ad hoc parameter choices (c, eps).
>
> The corollary holds for all choices of $c, \eps$. The values mentioned in the paragraph are provided solely as a concrete instantiation to help readers visualize the trade-offs in a concrete setting. That is to say, the corollary is the formal, convincing statement; the paragraph that follows is there to help the reader the formal statement.
>
> > Why would translators be different for paragraph-level (f) and segment-level (\phi)? Esp. with LLMs one would expect that they are the same.
>
> Adding more context (from segment to paragraph) will improve translation. Put differently, segment-level translation will be inherently limited, even with perfect translation. As a minimal example, consider the paragraph "I went to the bank today. The river was higher than yesterday." translated into French. A segment-level translator may translate bank→banque, whereas a paragraph-level translator will correctly translate bank→rive.
>
> > Why is the date of June 1, 2024 chosen?
>
> This is the knowledge cutoff date for pre-GPT-5 models. See our response to reviewer FqGT for a rigorous statistical analysis refuting a data contamination concern.
>
> > "prior work has validated": a reference would be nice here.
>
> Thank you for this suggestion, we have added the reference.
>
> > How is 99.9% estimated from 100 examples?
>
> For each of the 100 examples, 10 binary comparisons were made, one for each of 10 random permutations. Thus, the effective number of samples is $N=1000$.
>
> > Could you expand?
>
>  - Computational Complexity: Even if observational data is statistically sufficient to identify the translator, the computational problem of finding that translator (Empirical Risk Minimization) can be intractable for certain complex function classes, whereas interactive queries can sometimes simplify the search landscape.
>
>  - Fine-grained Delineations: If two competing translators differ only on very rare events (a "needle in a haystack" scenario), random observational sampling might take too long to uncover the difference, whereas interaction allows the learner to target that specific ambiguity directly.
>
>  - Out-of-Distribution Content: Observational learning guarantees performance only on the natural distribution of the data; if we require the translator to function on novel scenarios or behaviors that have not yet occurred naturally, interaction is necessary to generate these specific inputs.
>
>  - Counterfactuals: Observational data allows us to model correlations, but understanding the causal structure or testing hypothetical "what-if" scenarios (e.g., "how would the animal respond if the signal were slightly different?") requires the intervention provided by interaction."

---

### Official Review · Reviewer_JJun · 2025-11-01

**Soundness:** 3
**Presentation:** 3
**Contribution:** 2
**Rating:** 4
**Confidence:** 4

**Summary:**

This paper proposes an unsupervised evaluation for machine translation in a relatively high error stage. The motivation is to extend MT and its evaluation to animal language, where `interactive testing` as they call is, is more expensive or sometimes even infeasible with current resources.
The core contribution is a formal motivation and simulated experiments where ShufflEval is deployed as unsupervised MT metric.

**Strengths:**

- Unusual topic, adds diversity.
- Formal derivations and proof of the approach.

**Weaknesses:**

- Title is misleading. The motivation stems from animal sound translation, but the paper does not actually perform any experiments with whales.
- Artificial language setups might not be appropriately mimicking working with animals.
- Novelty: ShufflEval is not new and this evaluation doesn't add much to the adoption or success of it.
- Interesting discussion on the trade-off between cost and interactivity of feedback.

**Questions:**

- Why did you not use actual parallel, sentence-aligned data for the simulated? It would have removed some of the confounding factors/challenges.
- The "whalebreak" term is fun, but I'm not entirely sure if I understand it correctly? What does it entail?
- Can you motivate your data generation protocol?
- What other simple MTQE could you apply?

---

> ### Author Response · Authors · 2025-11-16
> **Response to Reviewer JJun**
>
> [1/2] We thank the reviewer for the thoughtful and constructive feedback.
>
> **Regarding the novelty criticism:**
>
> > ShufflEval is not new and this evaluation doesn't add much to the adoption or success of it.
>
> We respectfully disagree and would like to clarify the novelty. Shuffle-based tests have indeed been used to evaluate **monolingual** local coherence but have not been used for reference-free MTQE (RFQE). Our application of ShufflEval is to **segmentwise translations**, so it probes inter-segment semantic dependencies rather than just local fluency. In addition, we use ShufflEval specifically to address a failure mode of existing RFQE methods: they can over-score fluent **hallucinations** when no reference is available. Adapting a classic tool to a very different evaluation regime, and showing both theoretically and empirically that it works in this high-error, reference-free setting, is precisely the kind of “new application of an old idea” that has often been considered a meaningful contribution, e.g., applying Transformers to images.
>
> > Title is misleading. The motivation stems from animal sound translation, but the paper does not actually perform any experiments with whales.
>
> We are flexible on the title and appreciate this concern. The current framing reflects that the theory is genuinely about animal translation: for example, the “whalebreak” model explicitly compares observational data to interactive “playback experiments” and incorporates grounding such as animal location and environment. We also deliberately restrict our claims to animal communication; for human languages, it often makes more sense to rely on parallel data, and we do not wish to overclaim usefulness for low-resource human MT beyond the proof-of-concept role. To make this clearer, we are happy to adopt a more explicit title such as *“On Non-interactive Evaluation of Animal Communication Translators: A Proof-of-Concept via Human and Constructed Languages”* and to add a sentence in the introduction stating that all experiments are on human and constructed languages used as proxies, not on animals themselves.
>
> > Artificial language setups might not mimic work with animals.
>
> We agree that conlangs are not animal communication and will state this even more explicitly in the limitations. Their role in the paper is narrower: they provide a controlled way to stress-test the metric under extreme domain shift while retaining access to ground truth references. This lets us probe whether ShufflEval still correlates with reference-based evaluation in a regime where the source is very unlike typical human text, which is closer in spirit (though not in detail) to the animal setting.
>
> > Why did you not use actual parallel, sentence-aligned data for the simulated? It would have removed some of the confounding factors/challenges.
>
> For the conlangs, we do use fully sentence-aligned parallel data. The Wikipedia data are different: here, the “parallel” is at the article level, not exact sentence alignment. We considered several options for low-resource parallel corpora, but for the very rare languages we target, existing parallel datasets are both limited in size and at high risk of being “contaminated” by inclusion in the training data of current LMs. We therefore chose Wikipedia articles created after the model training cutoff, which gives us fresher and more genuinely low-resource material at the cost of imperfect alignment. This does introduce noise, but the fact that it works in this noisy setting is in some sense evidence of robustness. We will clarify this trade-off.
>
> > The “whalebreak” term is fun, but I'm not entirely sure if I understand it correctly. What does it entail?
>
> “Whalebreak” denotes an idealized interaction model in which each animal experiment is extremely informative: conceptually, one could imagine pointing at something and asking the whale “what is the word/coda for this?”, and each such interaction reveals $b$ bits of information about the translator, potentially the most useful $b$ bits. Real playback experiments are, of course, expensive, noisy, and hard to interpret. We deliberately analyze this optimistic interactive model because our claim is that non-interactive, observation-based evaluation can already be competitive in the high-error regime. If even such an idealized interactive model does not dominate observations, then the case for using costly real interaction becomes weaker. We will add a brief explanation of this to the main text when the term is first introduced.

---

> > ### Author Response · Authors · 2025-11-16
> > **[2/2]**
> >
> > [2/2]
> > > Can you motivate your data generation protocol?
> >
> > Our multi-stage conlang generation protocol is designed to ensure four properties: (i) diversity, by prompting the model to generate 10 languages at once, which reduces repetition; (ii) coherent concultures, so that the texts have rich, consistent semantics rather than being arbitrary strings; (iii) complete linguistic specifications, so that nontrivial syntax and lexicon can be used in the test texts; and (iv) systematic variation across the 10 conlangs, as summarized in Table 1. This makes the conlangs a useful stress-test family for a metric that is supposed to remain informative under large domain shifts. We will expand this motivation in Section 3.
> >
> > > What other simple MTQE could you apply?
> >
> > In the animal setting we target, there are no reference translations and no existing “ground truth” translators. In that regime, the only  reference free MTQE alternative reduces to judging “how good it looks” in the target language, which is exactly the situation in which hallucinations are most dangerous: a model can output fluent but entirely fabricated “translations” that score very highly on such reference-free judgements. In the paper, we therefore compare ShufflEval against a standard reference-based MTQE baseline (GPT-4 scoring, following Kocmi & Federmann 2023), which is the de facto evaluation standard where references exist.
> >
> > There are indeed other RFQE metrics such as COMET‑QE or related LM-based scores, but these are strongly targeted at failures in translation of human languages, not hallucinations
> >
> > **Score**: If we change the title, and in light of the the novelty and difficulty of making theoretical and empirical progress on animal translation in the absence of substantive animal translators, we ask if you might consider raising your score.

---

> > > ### Comment · Reviewer_JJun · 2025-11-24
> > >
> > > Thank you for your explanations and clarifications. I appreciate the updated title and presentation of scope. I'm updating my scope but would appreciate taking into account the below:
> > >
> > > Regarding human low-resource languages, the treatment of languages requires special care. For some of the languages chosen in this work, there have been dedicated efforts of creating parallel resources for MT (e.g. Tamazight (Alp Oktem, Mohamed Aymane Farhi, Brahim Essaidi, Naceur Jabouja, and Farida Boudichat. 2025. Correcting the Tamazight Portions of FLORES+ and OLDI Seed Datasets. In Proceedings of the Tenth Conference on Machine Translation, pages 1072–1080, Suzhou, China. Association for Computational Linguistics.), Khmer (WAT 2020 Khmer-English Parallel Data for the translation task)). It would be good to credit their existence when choosing not to work with them --- It would be a terrible misunderstanding to have them understood as zero-resource languages or on the same level of "unknown/understudied" as animal communication. Furthermore, Wikipedia articles of these languages are likely problematic in terms of quality as well (bots, translations, boilerplates, niche topics). This would be good to inspect and discuss (Google Translate is available e.g. for Tamazight, Sanskrit, Tulu, Khmer, etc and can be used for back-translation).

---

> > > > ### Author Response · Authors · 2025-12-03
> > > > **Thank you**
> > > >
> > > > We have added the following acknowledgment the dedicated high-quality datasets for some of the low-resource languages.
> > > >
> > > > > High-quality parallel datasets only exist for some of these languages, e.g., Tamazight \citep{oktem-etal-2025-correcting}. Therefore we use Wikipedia for a uniform approach.
> > > >
> > > > We agree adding that improves the paper. Thank you for the reference. We completely agree with your point about avoiding misunderstanding, see L501:
> > > >
> > > > > We also recognize that many low-resource human languages are associated with marginalized communities. Our use of these languages is strictly methodological: they provide a controlled setting
> > > > where ground-truth translations exist, allowing us to compare our reference-free metric against established evaluation methods. Low-resource languages were used in this paper because they are
> > > > (by definition) data-scarce—and often underrepresented in the large language models used in our
> > > > experiments—yet still come with parallel corpora that enables comparison to the gold-standard evaluation method (which uses references). Importantly, our use does not imply any similarity between
> > > > real human languages and animal communication.

---

### Official Review · Reviewer_wFLY · 2025-11-04

**Soundness:** 2
**Presentation:** 3
**Contribution:** 2
**Rating:** 2
**Confidence:** 4

**Summary:**

The authors propose a translation quality metric that is purely candidate-based, meaning it has no access to the source or any reference translations. Their metric, which they call ShufflEval, compares the coherence of translating the source as-is versus cutting the source into smaller segments and translating the permuted segments. Naturally, one would expect the original source to have much higher coherence.

The authors justify their proposed metric by appealing to basic statistical learning theory. They argue that although active learning/interactive systems can learn better models from less data than systems trained purely on observational data, this advantage essentially vanishes when collecting observational data is much cheaper than active learning, as one can just train systems on a lot more observational data. This is good news for the author's metric, as it relies solely on observational data.

Finally, the authors conduct experiments on low-resource human languages as well as so-called conlangs, which are artificial languages that they fabricated via prompting some powerful language model.

**Strengths:**

Overall, I found the original premise of the paper really intriguing and refreshing. I commend the authors on the first sentence of the abstract; it immediately drew me in and made me want to know everything about their work. I'm also a big fan of the terminology, e.g., the "whalebreak" model.

Though the applications of the authors' method seem a bit hypothetical or far-fetched at present, I found it relevant and interesting even as a purely thought-experiment.

**Weaknesses:**

I have two main issues with the paper: one regarding the theory/theoretical framing and one regarding the methodology.

First, the authors appeal to a fairly basic PAC bound (Eq 3) to argue that the expected risk of a system trained via empirical risk minimisation with respect to some loss $\ell$ will be close to the expected risk of a system trained via active learning. The result itself is fine, but using it to justify the use of the ShufflEval loss (defined in the equation in Section 2.4) is unsound for two reasons:
 1. It is easy to construct a system that minimises $\ell_{ShufflEval}$, which I call the "independent natural translator": it either ignores its input completely and outputs a piece of fluent English text, or (to make things more interesting) hashes the input and uses the hash as a seed to a random number generator to sample a piece of fluent English text from some dataset (e.g. English Wikipedia). As such, Eq 3 is vacuous for $\ell_{ShufflEval}$, and I cannot immediately see how the function class can be restricted to exclude these examples.
 2. This framing doesn't address the real problem: that we don't have reference translations. Given the story the authors tell in the rest of the paper, I would have expected Eq. 3 to connect the shufflEval loss to a loss that incorporates reference translations. As such, the authors should at least make it explicit that this is not what Eq. 3 represents.

As a more minor point, I would prefer the authors model translators as conditional distributions rather than functions, since in almost all cases sources have multiple valid translations. (Though this should not change the theory much)

My methodological issue has two parts also. First, I was disappointed that, despite the paper's incredible opening sentence and the careful ethics discussion in the introduction and at the end of the paper, there are no experiments on animal-to-English translation. Given this situation, I would either reduce the emphasis on animal translation in the main text (it occupies over one page!) or include some actual animal translation experiments.

Second, regarding the conlangs examples, the authors state: "As a result, one might expect our conlangs to be less human-like, which serves the purpose of stress-testing ShuffleEval beyond human languages." However, I randomly spot-checked the translations generated by language models in the supplementary material and found that most seem to produce excellent translations. As such, this calls into question whether these experiments are meaningful in the first place. At any rate, the author's statement above certainly is not borne out by this observation.

If the authors can elucidate if and how my reasoning is incorrect and address my concerns, I'll be happy to raise my score. However, if they find my concerns valid, then I'm afraid that I cannot recommend acceptance without significant modifications to the paper.

Miscellaneous:
 - page 5: $\rho(T, T')$ should be $\rho(T, T') = 0$
 - "We use LMs for several purposes, increasingly common practices in MT (Bavaresco et al., 2025), including" -- needs fixing

**Questions:**

How do the authors propose to pronounce their method?
Shuff - LEE - val, Shuffle - Eval or ShuffL - EE - val or some other way?

Does the authors' method have some connection with minimum Bayes risk decoding?

---

> ### Author Response · Authors · 2025-11-13
> **Clarification on the independent natural translator**
>
> Thank you for your thoughtful review. Before we respond in full, could you help us confirm what you intend by the “independent natural translator” (INT) counterexample with respect to our function class for ShufflEval?
>
> In the first version you mentioned, INT ignores the input and outputs some random fluent English text for each segment. Its loss is 0.5, which is large and would beat by a good translator.
>
> For the second version mentioned, we don't understand how it is a segmentwise translator meaning
> $$f(s_1 s_2 \ldots s_k) =  \varphi(s_1) \varphi(s_2) \ldots \varphi(s_k).$$
> It seems that INT outputs some fixed fluent English text consisting of the same distinct $k$ paragraphs $e_1 e_2 \ldots e_k$ regardless of the source. How can this be done with a segment-wise translation? For example, if $s_1=s_2=$"hello" in one example, then it would seem that $e_1=e_2$? More generally, how does $\varphi$ compute the exact index of a source paragraph across all translations?

---

> > ### Comment · Reviewer_wFLY · 2025-11-14
> >
> > Dear authors,
> >
> > Thank you for your immediate response. From your comment, my interpretation is that you might have understood my suggestion for the two variants of INT (approximately) the other way around.
> >
> > > In the first version you mentioned, INT ignores the input and outputs some random fluent English text for each segment.
> >
> > My understanding is that in your framework “outputting random fluent English text” is not possible, because you are modeling the translator as a function, not as a conditional distribution. So the first variant I was suggesting always just outputs the same sentence for any input.
> >
> > Having re-checked Section 2.4, I realise that this suggestion would need the segment translator $\varphi$ to be the constant function. When I was writing my review, I was imagining $f$ to be constant, but I now realise that this wouldn't necessarily be be a segment-wise translator.
> >
> > > For the second version mentioned...
> >
> > To be clear, my second suggestion was an attempt to model a translator that outputs random sentences, so to ensure segment-wise-ness, it would have to be $\varphi$ that hashes its input and outputs a pseudo-randomly picked English sentence based on the hash.
> >
> > > More generally, how does $\varphi$ compute the exact index of a source paragraph across all translations?
> >
> > I don’t understand this question, but hopefully my response above helps clear things up?
> >
> >
> > Taking a step back, I maintain that you should (at least in some follow-up work) consider stochastic translators, as they are a better model for translators.
> >
> > >  Its loss is 0.5, which is large and would beat by a good translator.
> >
> > This is a good point, and indeed you mention this possibility in the introduction. However, I think this raises a new point of confusion, which I overlooked in my review: my understanding is that we would hope that an ideal system *maximises* $\ell\_{ShufflEval}$. As such, it is somewhat strange to call it a loss and I am also less clear on what the your intended argument is when evoking Corollary 2: do they mean to emphasize that the loss of non-interactive systems *lower-bounds* the loss of interactive systems? If yes, then I must admit, I completely missed this point and would recommend that you make it more explicit.
> >
> > Also, to provide some context that might be helpful: I took the idea of independent natural translator from the recent work of Flamich et al. (2025), which studies the interaction of reference and reference-free metrics (they call the former “accuracy” and the latter “naturalness” metrics). In my understanding, the relevant part of the paper I mention is that accuracy (semantic correctness of translation) and naturalness (how good the translation sounds) ought to be assessed together, as there is a tradeoff between them. My impression after reading your paper was that ShufflEval would be an instance of a “naturalness” metric, but you didn't mention how it relates to some notion of accuracy, which left me wondering how your work fits into the broader picture of the accuracy-naturalness tradeoff.
> >
> > ## Reference
> >
> > Flamich, G., Vilar, D., Peter, J.-T., and Freitag, M. (2025). You cannot feed two birds with one score: the accuracy-naturalness tradeoff in translation. In Second Conference on Language Modeling.

---

> ### Author Response · Authors · 2025-11-21
>
> Thank you so much for clarifying.
>
> > Taking a step back, I maintain that you should (at least in some follow-up work) consider stochastic translators, as they are a better model for translators.
>
> We added the following remark: “For simplicity, we assume translators are deterministic (though the analysis remains essentially unchanged for randomized translators)”. The only change would be that we would have to take expectations over the translations everywhere.
>
> > my understanding is that we would hope that an ideal system maximises $\ell_\text{ShuffleVale}$
>
> Thank you for pointing out this confusion. In the theory section, we use loss which (as standard) is better when minimized. In the experiments and discussions, we use the shuffleval *score* which, like accuracy, is better when larger. To address this confusion, we’ve added:
> “In experiments, we consider the ShufflEval score $=1-\ell_{\mathrm{ShufflEval}}(T)$ which is better when larger.”
>
> Regarding the **accuracy-naturalness tradeoff**, in fact ShufflEval is actually much more about accuracy. It is well-suited for detecting hallucinations which are often very natural but completely inaccurate. As we say in the paper:
> > ShufflEval complements these other RFQEs in that it is more resilient to hallucinations, but it is not a replacement. For example, it may be insensitive to AlTeRnAtInG cAsE, which impedes readability without significantly altering meaning. Thus, ShufflEval can be combined with other RFQE metrics.
>
> Our example of alternating case illustrates how ShufflEval is a poor measure of naturalness. We are adding a link to the (Flamich et al, 2025) reference you mentioned in this part for further reading.
>
> We will respond to the rest of your initial review shortly.

---

> > ### Author Response · Authors · 2025-11-22
> > **Response to the rest of the review**
> >
> > Did the above discussion address your concerns regarding the theoretical analysis? Back to the rest of the review.
> >
> > We appreciate your engagement and are pleased that there appears to be general agreement that Animal-to-English translation is a problem of profound scientific interest. We would like to further clarify our perspective on the framing and the space devoted to this topic.
> >
> > >  I would either reduce the emphasis on animal translation in the main text (it occupies over one page!) or include some actual animal translation experiments.
> >
> > We would be grateful for any specific suggestions regarding animal translation experiments that could realistically be performed with the data currently available. At present, large-scale efforts to collect such data are ongoing, including initiatives such as Project CETI. However, until sufficiently rich data exists, we believe that genuine animal-to-English translation experiments for complex animal communication systems are not yet feasible, especially given the data-hungry nature of current Transformer-based language models.
> > Reducing the emphasis on animal translation would, in our view, significantly weaken the paper, as this is the primary motivation for the work. Without this framing, both the theory and experiments would lose much of their context and purpose. For low-resource human languages or constructed languages, our approach is less clearly justified, as acquiring parallel data is generally more feasible and would likely be more effective. We therefore devote space to explaining why our theory and experiments are relevant to animal translation in particular. One important contribution of the work is that it demonstrates how evaluation methods for potential animal translators can be explored even before substantial real-world animal translators exist. While we acknowledge that our current experiments are not conclusively tied to animal translation, we believe the direction shows sufficient promise to justify exploration.
> >
> > >  the authors state: "As a result, one might expect our conlangs to be less human-like, which serves the purpose of stress-testing ShuffleEval beyond human languages." However, I randomly spot-checked the translations generated by language models in the supplementary material and found that most seem to produce excellent translations… the author's statement above certainly is not borne out by this observation.
> >
> > We would appreciate clarification on what is meant by “excellent translations” and why this is seen as contradicting the claim that the conlangs are less human-like. Our intention is to create languages that diverge from typical human linguistic patterns, thereby stress-testing the evaluation methodology.
> >
> > To be concrete, here is the target “gold” translation of the first example (Avaru.json) from the supplementary material.
> >
> > ```
> > We play Spiral at night.
> > Beads glow and fog drifts in Iqan’s light.
> > The two moons pull, and the wind gains a slow beat.
> > Do not make a hard knot, will you?
> > We offer Depth, even when someone is Shallow.
> > The beads complete passes, and Sum is carried to the center.
> > ```
> >
> > Compared to English Wikipedia articles, this text exhibits significantly less typical human-like structure and content. They are, of course, written in English because they represent ground-truth translations. The fact that some models produced translations close to this is expected, since as stated in the opening, we are trying to design algorithms now to validate a future where we have a plausible animal translator. (Moreover, such evaluations may useful to learn to translate in the first place.)
> >
> > Thank you for pointing out the miscellaneous typographical issues. We will correct these in the next revision.
> >
> > Questions:
> > > How do the authors propose to pronounce their method? Shuff - LEE - val, Shuffle - Eval or ShuffL - EE - val or some other way?
> >
> > SHUFF-leval.
> >
> > > Does the authors' method have some connection with minimum Bayes risk decoding?
> >
> > We analyze learning a translator $f \in \mathcal{F}$ by minimizing expected loss $E[\ell(f(S),Z)]$ over a data distribution, which is slightly similar in spirit. If one had a posterior over $Z \times S$ one could try actual Bayes risk decoding.

---

> > > ### Comment · Reviewer_wFLY · 2025-11-26
> > >
> > > Dear authors,
> > >
> > > Thank you for your response.
> > >
> > > > Regarding the accuracy-naturalness tradeoff, in fact ShufflEval is actually much more about accuracy.
> > >
> > > I understand that this is your claim. However, since the ShifflEval score/loss only has access to the candidate translations, I don’t see how it is formally connected to some notion of accuracy, i.e., a metric that is allowed to depend on the source text/data or reference translations.
> > >
> > > My understanding is that ShufflEval is essentially an exchangeability test, using the very sensible assumption that the segments should not be exchangeable. However, it is not clear how identifying that a translation is indeed non-exchangeable (good ShufflEval performance) ought to imply good accuracy. Is there some much simplified setting in which such a claim could be shown formally?
> > >
> > > > RFQEs
> > >
> > > My understanding is that RFQE metrics are still allowed to depend on the source segment, whereas ShufflEval is not. Or do you have something else in mind?
> > >
> > > > We would be grateful for any specific suggestions regarding animal translation experiments
> > >
> > > While I certainly see and agree with you that reducing the focus on animal translation would significantly weaken the paper, I hope you also see my point that a paper with animal translation in the title and extensive discussion of animal translation, but without any explanation of why there are no animal experiments, isn’t viable.
> > >
> > > Unfortunately, I don’t have any suggestions for animal data. I presumed, given that you cited three such studies in the first sentence of the introduction, that there was some open data set. If this is not possible, including a variant of your explanation for the lack of data in the paper would be good, as it clarifies your motivation and the current situation a lot. I think it would be a reasonable compromise.
> > >
> > > > We would appreciate clarification on what is meant by “excellent translations”
> > >
> > > Taking the first example from `Avaru.json`, the first target sentence is `We play Spiral at night.` Checking the translations produced by the LLMs, I see that `4.1` produced the translation `At dusk, we play Spiral together.` which is an almost perfect translation. My understanding is that the source sentence for this example is `qo nari=B muru {shuffle-low-long-between} zul-n =oth.` Given that only thing the LLMs saw other than the source is the conlang/conculture descriptions, my hunch is that they are leaking too much information.

---

> ### Author Response · Authors · 2025-11-29
>
> > since the ShifflEval score/loss only has access to the candidate translations, I don’t see how it is formally connected to some notion of accuracy, i.e., a metric that is allowed to depend on the source text/data or reference translations.
>
> This connection is justified empirically, not formally, which is the whole purpose of the experiments in the paper.
>
> > Checking the translations produced by the LLMs, I see that 4.1 produced ... an almost perfect translation... Given that only thing the LLMs saw other than the source is the conlang/conculture descriptions, my hunch is that they are leaking too much information.
>
> In the conlang experiments, the language model translators are given the whole description of the conlang (many thousands of tokens) as reference material in order to facilitate translation. This is stated on line 459: "Second, in order to translate the conlang text, the conlang and conculture definitions were provided in the LM prompt, as seen in Figure 8 (bottom)."
>
> > My understanding is that RFQE metrics are still allowed to depend on the source segment
>
> Thank you for pointing out this possible confusion. We have clarified by adding "We use reference-free to mean no reference translations; most MTQE uses the source text, whereas our metric is target-only (it uses only translated segments plus their temporal order)."
>
> > Unfortunately, I don’t have any suggestions for animal data.... a variant of your explanation for the lack of data in the paper would be good
>
> We have added this sentence to the document "there is currently inadequate animal data to translate complex animal communication systems". We agree that it worth stating this explicitly.

---

### Author Response · Authors · 2025-12-03
**AC: Scores should've be significantly raised**

**ACs**: during the rebuttal, you can see that several of the ratings were going to be raised. For example, wFLY who gave a rating of 2, acknowledged several misunderstanding in their reading of the paper, both in the theory and experimental sections.

This paper introduces a new problem. Unlike papers that make incremental progress but are easy to evaluate, such papers often require iteration during the rebuttal phase to ensure adequate reviewing. The rebuttal phase was successful in this respect, except for the fact that the final scores could not be raise by reviewers.

We have updated the paper with changes marked in red to help clarify several confusions pointed out by reviewers. We are grateful to the reviewers and area chairs, thank you.

---

### Meta-Review · Area_Chair_Jzee · 2026-01-03

**Summary:**

This paper introduces a new evaluation method for machine translation, which is an instance of machine translation quality evaluation without any reference translation available. The authors use ShuffleEval to define the evaluation metric, and the evaluation is conducted by randomly shuffle the segments. Through studies and human created language experiments, the authors show the value of the proposed evaluation method.

The paper positions at a perspective paper that the authors use a whale-to-english translator as an imitation, the reviewers raise concerns about the evaluation definition, the evaluation data size, and also the evaluation models. Though the paper is interesting, these concerns are hard to give satisfied rebuttal. Especially about the experiments about animals...(though it's not possible for authors to conduct at this stage)

**Reviewer Concerns:**

The paper positions at a perspective paper that the authors use a whale-to-english translator as an imitation, the reviewers raise concerns about the evaluation definition, the evaluation data size, and also the evaluation models. Though the paper is interesting, these concerns are hard to give satisfied rebuttal.

Some other concerns are solved such as the unclear definition, some simple experimental questions.

**Reviewer Scores:**

Reviewer wFLY and JJun are somehow not possible to change their evaluation according to the current review and the provided rebuttal, which are still negative view.

---

### Decision · Program_Chairs · 2026-01-26

Reject